# Adaptation to novel spatially-structured environments is driven by the capsule and alters virulence-associated traits

Amandine Nucci [1], Eduardo P. C. Rocha [1] & Olaya Rendueles [1] ✉

The extracellular capsule is a major virulence factor, but its ubiquity in free-living bacteria with large environmental breadths suggests that it shapes adaptation to novel niches. Yet, how it does so, remains unexplored. Here, we evolve three *Klebsiella* strains and their capsule mutants in parallel. Their comparison reveals different phenotypic and genotypic evolutionary changes that alter virulence-associated traits. Non-capsulated populations accumulate mutations that reduce exopolysaccharide production and increase biofilm formation and yield, whereas most capsulated populations become hyper-mucoviscous, a signature of hypervirulence. Hence, adaptation to novel environments primarily occurs by fine-tuning expression of the capsular locus. The same evolutionary conditions selecting for mutations in the capsular gene *wzc* leading to hypermucoviscosity also result in increased susceptibility to antibiotics by mutations in the *ramA* regulon. This implies that general adaptive processes outside the host can affect capsule evolution and its role in virulence and infection outcomes may be a by-product of such adaptation.

Evolution experiments enable the testing of keystone questions of evolutionary theory[1–3], revealing the mechanisms by which populations adapt[4–7], how latent phenotypes, that are not directly selected for, evolve[8,9], and even forecasting evolutionary outcomes[10]. Numerous studies show the central role of genetic background and nutrient sources in shaping adaptation to novel environments. Yet, few studies address how traits core to the physiology and behaviour of a species generally impact the evolutionary trajectories at the genotypic and phenotypic level[11]. This is especially crucial when studying how components at the cell surface which directly respond to environmental cues and stresses, such as the bacterial capsule, affect bacterial microevolution[12].

When present, the bacterial capsule is the first cellular structure in contact with the environment. It is mostly studied for its role as an important virulence factor, as it masks surface antigens and limits the immune response[13,14]. However, most bacteria with capsules are free-living and not associated with a host, suggesting that virulence could be a by-product of adaptation independent of the host[15]. For instance, the capsule increases fitness by limiting co-colonisation of competitors[16] and counteracting Type VI secretion system-mediated

bacterial attacks[17]. Furthermore, bacteria encoding capsules are genetically more diverse, carry more mobile genetic elements and have fast-evolving gene repertoires[18]. This could result in a higher potential for adaptation. Accordingly, our previous metagenomic analyses revealed that these bacteria have larger environmental breadths, that is, they occupy more ecological niches than non-capsulated bacteria[15]. This is also true for many opportunistic pathogens, which are capsulated, and are known to survive in a wide range of ecosystems. For instance, the enterobacterium *Klebsiella pneumoniae* is a gut commensal of diverse animals, but it is also found in association with plants or in aquatic environments[19]. It is also becoming an increasingly important nosocomial multidrug resistant human pathogen[19]. Previous research showed that non-capsulated clones can rapidly emerge and have large fitness advantages in well-mixed nutrient-rich environments or in populations under phage pressure, as evidenced by short evolution experiments[20–22]. Genomic studies also determined that at least 3.5% of *K. pneumoniae* are non-capsulated[23]. These variants exist across the different lineages of *K. pneumoniae* and tend to be very recent, suggesting that at a later stage, they are either counter-selected or complemented by recombination[23]. Finally, the

[1]Institut Pasteur, Université de Paris, CNRS, UMR3525, Microbial Evolutionary Genomics, F-75015 Paris, France. ✉e-mail: olaya.rendueles-garcia@pasteur.fr

capsule in *K. pneumoniae* strongly influences the frequency and nature of horizontal gene transfer, which also affects the long-term evolution of the species[23].

However, whether the presence or absence of the capsule affects the first steps of adaptation to novel environments and whether it determines evolutionary outcomes, to our knowledge, has not directly been tested. To do so, we set up an evolution experiment for *ca.* 675 generations (102 days). To reduce the effect of strain and environmental specificities, we study three phylogenetically-distant strains in five environments varying in nutrient sources and carrying capacities. More precisely, we evolve six replicate populations of two different hypervirulent *K. pneumoniae* (Kpn BJ1 and Kpn NTUH) belonging to the two most widespread capsule serotypes (K2 and K1, respectively) and one environmental *K. variicola* strain (Kva 342) (Table 1[24–28]). Their three respective isogenic non-capsulated mutants are also propagated in parallel. The populations evolve in environments relevant to *Klebsiella* physiology, such as artificial sputum medium (ASM)[29], artificial urine medium (AUM)[30] that mimic host-related nutritional conditions, filtered potting soil, and two typical laboratory media with different levels of nutrients: rich Luria–Bertani medium (LB) and minimal medium supplemented with 0.2% of glucose (M02). Populations are grown in 2 mL in 24-welled microtiter plates without shaking, allowing for spatial structure. The latter recapitulates better the complexity of natural environments and allows for the emergence of more diverse adaptation mechanisms[31], compared to well-mixed environments, due to less selection pressure on faster growth rate and the possibility to colonise different micro niches.

We hypothesise that the diversity in adaptation mechanisms in structured environments could also be contingent on the presence or absence of the capsule and the nutrient content of the media[32]. Specifically, in nutrient-rich conditions, where non-capsulated cells easily emerge both in well-mixed and static environments[32,33], we expect capsulated populations to inactivate the capsule or limit its expression. Non-capsulated populations could then adapt, as shown in previous experiments in static environments, by increased cell-to-cell interactions by virtue of adhesion factors like fimbriae, previously inhibited by the presence of the capsule[34]. This would result in the formation of large aggregates that fall to the bottom of the well[35]. On the contrary, in nutrient-poor conditions, the presence of the capsule is an important fitness determinant and increases group yield[32]. These populations should remain capsulated and could adapt by increasing capsule production to increase group productivity or by increasing viscosity. The latter results in the colonisation of the air-liquid interface by the formation of a bacterial mat, previously shown to be adaptive[36,37]. The increased capsule production in capsulated populations or increased cell-to-cell interactions in non-capsulated populations should impact biofilm formation and hypermucoviscosity, commonly associated with hypervirulence[38,39]. Thus, the generic process of adaptation to novel environments in *Klebsiella pneumoniae* outside the host could also impact infection outcomes.

Our experimental setup (Supplementary Data 1) allows us to specifically test capsule effects on the patterns of evolutionary change at both phenotypic and genotypic levels, whilst also testing for virulence-related by-products of adaptation. We also follow the emergence of diverse capsule phenotypes through time and show the diversification of the population and the coexistence of non-capsulated and mucoid colonies. Overall, our study highlights the capsule as a major driver of adaptation to novel environments and provides insights into how general adaptive processes influence capsule-associated traits relevant to human disease outcomes, such as biofilm formation and antibiotic resistance, in a major nosocomial pathogen. Finally, it further challenges the long-standing hypothesis that the capsule is maintained primarily as a response to the host or to other biotic pressures, and suggests that both environmental structure and nutrient amount can select for the capsule.

## Results

### Ancestral presence of capsule drives phenotypic evolution

After 102 days of evolution in different environments, we verified that end-point populations had adapted, that is, they were fitter than their ancestors. Direct competitions between randomly sampled end-point populations and antibiotic-marked ancestors, that displayed no growth defects, revealed that both capsulated and non-capsulated populations had a competitive advantage over their respective ancestors in the evolutionary environment (average fitness increase of 58 and 36%, respectively (Fig. S1)).

To precisely test whether capsulated and non-capsulated populations adapt by different mechanisms, we measured population yield, an adaptive trait, after 24 h, in all end-point populations in their respective evolutionary environments. We also tested whether populations increased the production of surface polysaccharides, increased viscosity, as measured by the hypermucoidy phenotype (HMP)), and increased biofilm formation. These three traits are also known to affect the virulence of *Klebsiella*, they are expected to be affected by the generic process of adaptation, and are dependent on the presence or absence of the capsule. Due to experimental constraints, colony-forming units (CFU) could not be analysed for populations with high HMP, mostly capsulated populations evolved in LB and ASM (see Methods). Biofilms and capsule production could not be measured in soil because they were below the limit of detection of the method. Finally, the HMP was measured in M02 for all populations, as the media strongly biases HMP measurements. For each of these four traits, we tested the direction and magnitude of change of each population relative to its ancestor (Table S1). Absolute ancestral values and relative values of each independent clone for each trait are provided in Fig. S1. The changes observed in all four traits were dependent on the ancestral strain (Multifactorial ANOVA, df = 2, $P < 0.001$, Table S2). As expected, post hoc analyses showed that there were significant differences across the three different strains. Biofilm production and CFU were also dependent on the environment, but changes in surface polysaccharide production or HMP were not. No significant differences were observed between host-mimicking environments (AUM and ASM) when compared to the others (LB, M02 and soil) for these two variables (Multifactorial ANOVA, df = 1, Biofilm $P = 0.9$; CFU $P = 0.1$).

More importantly, all four traits were dependent on the presence/absence of the capsule. Further, stepwise regressions showed that the capsule genotype was the primary predictor driving the changes in

## Table 1 | Details of the strains used in this study

| Strain | Species | Isolation | ST | K-serotype | O-serotype | *rmpA* | Genome size (MB) | # of IS | Ref |
|---|---|---|---|---|---|---|---|---|---|
| NTUH-K2044 | *K. pneumoniae* | Liver abscess, Taiwan | ST23 | K1 | O1v2 | 2 | 5.25 | 21 | 26 |
| BJ1 | *K. pneumoniae* | Liver abscess, France | ST380 | K2 | O1v1 | 1 | 5.26 | 10 | 27 |
| 342 | *K. variicola* | Maize, USA | ST146 | K30 | O3/O3a | 0 | 5.64 | 17 | 28 |

Kleborate[24] was used to determine the species, the sequence type (ST), K-serotype, O-serotype and presence of hypermucoidy genes (*rmpA*). Confidence values for serotyping and species assignation were classified as 'strong' or 'very high'. Insertion elements (IS) were detected using ISfinder[25].

biofilm formation and the second most important predictor of changes in surface polysaccharide production, population yield and HMV (Table S2B). More specifically, out of the 41 possible trait-environment comparisons between capsulated and non-capsulated populations, populations evolved in opposite directions 16 times (~40%), of which 12 significantly so (Table S1), suggesting that there is a significant effect of the capsule in driving the direction of phenotypic evolution. As we hypothesised, non-capsulated populations adapted by altering growth parameters, as revealed by increased yield (t-test, a difference from 0, $t = 6.4$, $N = 83$, $P < 0.001$, Fig. S2). Further, non-capsulated populations also displayed increased biofilm formation (t-test, difference from 0, $t = 3.8$, $N = 83$ populations, $P = 0.001$), suggesting enhanced cell-to-cell and cell-to-surface interactions. Unexpectedly, non-capsulated populations increased the production of surface-associated polysaccharides (t-test, difference from 0, $t = 5.1$, $N = 83$, $P < 0.001$). Opposite to non-capsulated populations, capsulated populations evolving in nutrient poor media did not adapt by increasing yield (Fig. 1). However, as anticipated, cells increased viscosity as revealed by increased HMP (t-test, a difference from 0, $t = -8.3$, $N = 81$, $P < 0.0001$). Further, this resulted in decreased biofilm formation (t-test, difference from 0, $t = -2.2$, $N = 81$, $P = 0.02$). Our results also show phenotypic diversification, similar to other populations evolving in static environments. Indeed, some capsulated populations dramatically reduced capsule production, whereas others remained near ancestral values or increased production. These general conclusions are also supported when analysing data in the two nutrient-poor environments for which all traits could be measured, namely AUM and M02 (Fig. 1), suggesting that the amount of nutrients available does not drive evolutionary outcomes as we had hypothesised.

We then analysed whether changes in adaptive or virulence-associated traits correlate positively or negatively and whether this could depend on the capsule (Fig. 1). Indeed, in capsulated genotypes, biofilm formation and capsule production show a strong negative correlation, but in non-capsulated populations, increased biofilm positively correlates with increased production of surface polysaccharides. Similarly, in populations derived from capsulated ancestors, higher levels of capsule production positively correlate with increased hypermucoidy, whereas, no correlation was observed between the increase in surface-associated polysaccharides and HMP in non-capsulated populations (Fig. 1).

Taken together, our results show that the presence of capsules in adapting populations strongly shapes the direction of phenotypic change and suggests that capsulated and non-capsulated populations may be adapting to structured environments by different mechanisms.

**The capsule shapes genotypic changes in evolving populations**
To determine the genetic basis of phenotypic adaptation, we sequenced one randomly-chosen clone per population. On average, we observed four mutations per clone, with significant differences in the number across strains and environments (multifactorial ANOVA, Strain ($F = 16.3$, df = 2), Environment ($F = 10.9$, df = 4), $P < 0.001$) (Fig. S3A). The presence of the capsule did not affect the total number of mutations per clone nor the number of different genes that acquired a mutation.

To test whether some cellular processes were preferentially targeted by mutations, we assigned a COG process category (Clusters of Orthologous Group) to each gene family in the ancestral genomes[40] and counted the proportion of mutations per COG relative to the whole genome (Fig. 2A). Patterns are similar in clones descending from capsulated and non-capsulated ancestors, but we observed opposite trends in defence mechanisms (V). Specifically, mutations in this group are over-represented in clones from the capsulated background, but are under-represented in non-capsulated clones. Direct comparison in the number of mutations within each COG group between capsulated and non-capsulated clones showed that the latter accumulate

significantly more mutations in genes associated with cell motility (N) (Fisher's test, Odds ratio 0.3, $P < 0.001$) than the capsulated clones. This is mostly driven by mutations found in the *mrkABCDF* operon, which codes the type 3 fimbriae[41]. These mutations accumulate significantly more in non-capsulated populations ($\chi^2$, $P < 0.001$). Capsulated clones are enriched in mutations involved in transcription (K) (Fisher's test, Odds ratio = 1.6, $P = 0.05$) and cell wall, membrane and envelope biogenesis (M) (Fisher's test, Odds ratio = 2.5, $P = 0.005$).

To precisely pinpoint the impact of the ancestral genetic background on mutations at the gene level, we established the pangenome of the three ancestral strains. There was very little overlap between the mutated genes in clones descending from capsulated and non-capsulated ancestors (Fig. 2B). Dissimilarity tests revealed that there are few mutations common to capsulated and non-capsulated clones descending from the same genetic background, comparable to the differences across strains (Table S3 and Fig. S3B). We, however, identified two operons that were consistently mutated in all strains: the abovementioned *mrkABCDF* operon and the capsule operon (Fig. S3C). Additionally, we also found numerous mutations in known capsule regulators. To systematically analyse capsule-related mutations (capsule operon and regulators), we compiled a list of 143 genes identified by mutagenesis to affect capsule production (either up or down-regulation, Supplementary Data 1, see Methods)[42,43] and checked whether mutations occurred in genes (or neighbouring intergenic regions) homologous to known capsule-related genes[42,43]. Mutations in capsule-related genes account for 19% of the total, yet capsule-related genes represent only ~1.9% of the genome (as calculated by the total nucleotide length of all such capsule-related genes in the reference genome of Kpn NTUH)[42,43]. This indicates that they are major targets of selection in all evolutionary treatments as they are extremely over-represented ($P < 0.001$ for deviation from the expectation of 18 mutations under a null assumption of random distribution of the 673 mutation events identified in the evolved clones, two-tailed binomial test). Interestingly, clones descending from capsulated ancestors mutated preferentially within the capsule operon, as already suggested by the COG analyses, whereas clones descending from non-capsulated ancestors had mutations in genes regulating capsule production (Fisher's test, Odds ratio = 18.86, $P < 0.001$) (Fig. 2C).

To test the effect of mutations in capsule regulatory genes, we restored capsule expression by *wcaJ* complementation in trans and quantified capsule production of several clones in the environments in which they evolved. We tested several deletions in the hypervirulent plasmid of NTUH, all of which resulted in the absence of a gene annotated as *rcsA_3*, a potential homolog of *rmpA* (40% identity). These mutations did not affect capsule production. We hypothesised that it could affect HMP, but this could not be tested as capsule complementation did not restore HMP (data not shown). This could be explained by the tight multifactorial regulation of HMP[44]. However, the mutations in capsule regulators found in the chromosome all significantly reduced the amount of capsule produced (Fig. 2D).

Our results show that the presence of a capsule strongly affects the first steps of adaptation to novel environments. Capsulated clones tend to have mutations in genes affecting directly capsule synthesis, whereas non-capsulated clones tend to have mutations in regulators to diminish, but not fully abolish, capsule production.

**Capsule inactivation emerges readily but rarely fixes in structured environments**
To test population diversification dynamics which readily emerge in static environments, notably in terms of capsule production, and whether evolution in nutrient-rich media over long periods of time results in capsule inactivation as previously observed[32,33], we periodically plated capsulated populations. We specifically tested whether the capsule could be maintained during hundreds of generations in the absence of biotic pressure. We plated all populations descending from

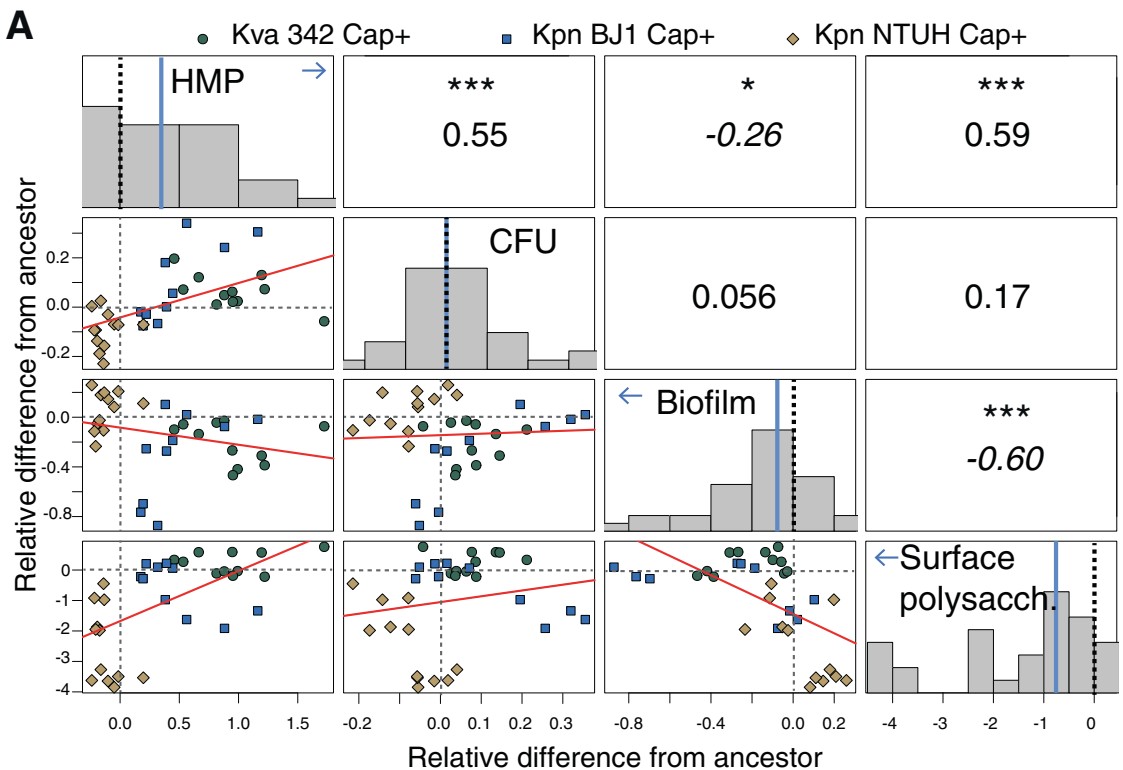

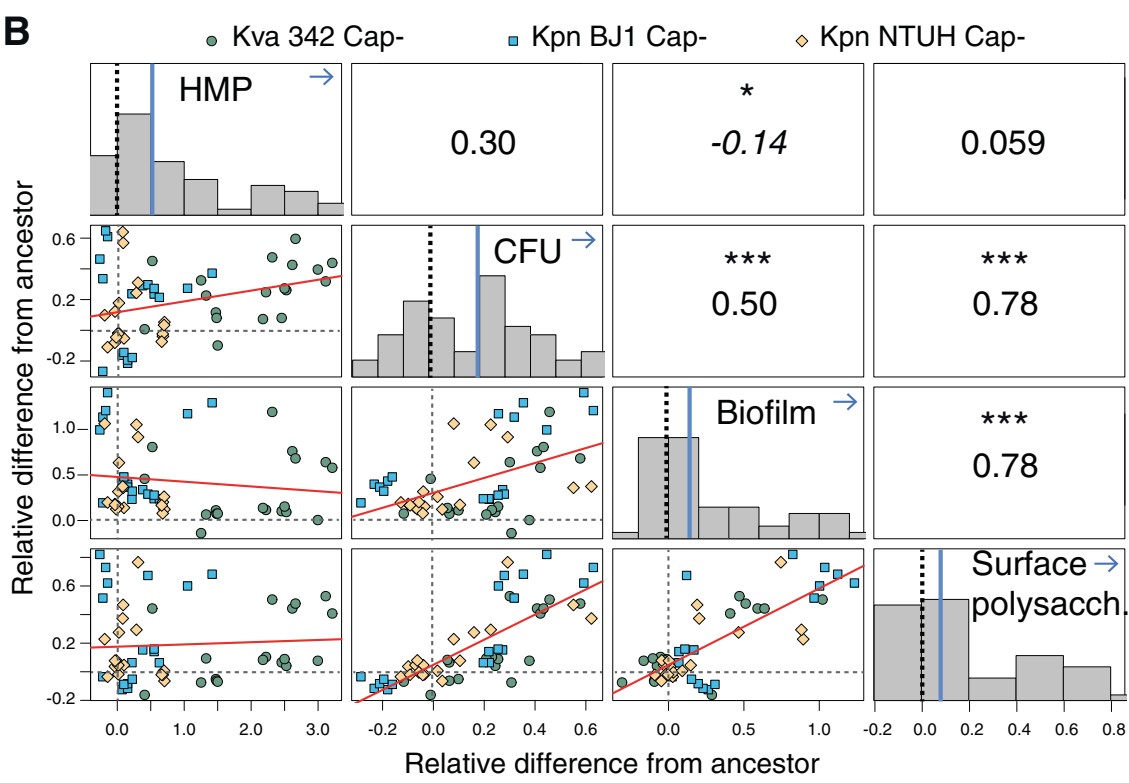

capsulated ancestors at regular intervals and visually examined the proportion of capsulated and non-capsulated clones (see Methods, Fig. S4). In ASM and LB, not all time points for all populations could be examined due to a very persistent HMP, which precluded dilution plating. We observed the emergence of non-capsulated clones in 61 independent populations (75%), but the dynamics differed across strains and environments (Fig. S4). In a majority of the populations,

non-capsulated clones emerged at very low frequencies and were rapidly outcompeted. In some populations, we observed periods of stable coexistence between capsulated and non-capsulated clones (Fig. S3, example: Population 6 of Kpn BJ1 or Population 5 of Kva 342 in ASM, and most populations of Kpn NTUH in AUM), but in most cases, the frequency of non-capsulated alleles changed rapidly. This resulted in the elimination of the non-capsulated clones from the population in

**Fig. 1 | Phenotypic changes are dependent on the presence and absence of the capsule.** Direction and magnitude of phenotypic change capsulated (**A**) and non-capsulated (**B**) populations relative to their respective ancestors. Dashed lines indicate ancestor baseline (0). The histograms represent the distribution of the difference between each population to their respective ancestor. The median is indicated in blue. Arrows highlight the direction of trait change, and statistics correspond to the $t$-test, a difference from 0, ($N$ = 36). Statistics reported in the text correspond to all populations from all environments. The scatterplots represent the correlation between changes in pairs of traits. Each dot represents the average of at least three biological replicates for each end-point population. Error bars are not shown for clarity purposes. Red line fits a linear model. The numbers represent two-sided Pearson's correlation. Negative correlations are highlighted in italics. *$P$ < 0.05; ***$P$ < 0.001. Only data from M02 and AUM are included. To avoid confusion, we redefined capsule production as surface-associated polysaccharides ('Surface polysacch') produced both by the non-capsulated and capsulated populations. HMP hypermucoid phenotype, CFU colony-forming units. Source data are provided as a Source Data file.

most cases. Indeed, at the end of the experiment, 67 populations out of 81 descending from a capsulated ancestor were still dominated by capsulated clones, and in 53 (65%) of them, all sampled individuals remained capsulated (Fig. 3). These populations were found in all environments and in all strains, and capsule maintenance was independent of both strain and environment (Two-way ANOVA, Strain ($F$ = 0.7, df = 2), Environment ($F$ = 0.6, df = 4), $P$ > 0.05). In only 7 out of 81 populations (9%), the capsulated clones were driven to extinction. In these populations, non-capsulated clones swept the population fast and steadily.

Our results suggest that capsule mutants emerge easily but rarely fix in structured environments, suggesting that they are either out-competed or their capsule restored by recombination with other clones in the population.

### The mutational mechanisms are strain-specific

The genetic basis of capsule inactivation in structured environments followed previously observed patterns, namely mutations in *wcaJ*, the first gene of the biosynthetic pathway[23,33]. Of the 81 randomly-chosen clones descending from a capsulated ancestor, many of which had mutations in the capsule operon, only 12 were non-capsulated. Non-capsulated clones of strain Kva 342 emerged primarily by IS insertions, small mobile genetic elements (-0.7–2.5 kb) that can vary in type and copy number across genomes, whereas those of Kpn BJ1 and Kpn NTUH resulted in changes in the reading frame or premature stop codon in *wcaJ, wzc* or *rfaH*, genes essential for capsule synthesis[45,46] (Table S4).

To test whether spatial structure influenced the mechanisms of capsule inactivation, we compared our results to those from a short evolution experiment, *ca* -20 generations, in well-mixed environments. This short evolution experiment included the three focal strains of this study[32] and was only partially analysed in ref. 23. We found that mutational mechanisms of capsule inactivation were not affected by the structure of the environment. For instance, all events of capsule inactivation in Kva 342 are due to IS insertions in both experiments (total of $N$ = 8), whereas the opposite is true for Kpn BJ1 for which no IS insertions were found in the capsule operon ($N$ = 8). For strain Kpn NTUH, a mix of IS insertions, non-synonymous mutations and single base-pair deletions were found in both experiments ($N$ = 11).

The differences observed across these three strains suggested that the mutation mechanisms could significantly vary across genetic backgrounds. To test this and verify whether mutation patterns in these three strains also occur in other *Klebsiella* strains, we analyzed 73 non-capsulated clones descending from 16 different capsulated strains from the aforementioned experiment in well-mixed environments[32]. Almost half (44%) of capsule inactivation events were due to an IS insertion (Fig. 4A), but this was strain-specific (Fig. S5A). Some strains, including Kva 342, mutate very frequently (or exclusively) by IS insertions, whilst others seldomly. There was no correlation between the total number of IS in a strain and the frequency of IS-dependent capsule inactivation (Fig. S5B). Capsules were mostly interrupted by IS903 from the IS5 family, which uses a replicative mechanism of transposition. Two other capsules were interrupted by IS from the IS91 and IS3 family, in multidrug-resistant NJ ST258 and Kpn ST2435, respectively. The presence and number of IS of these three families in the genome significantly correlated with the frequency of capsule inactivation by such elements (Spearman's rho = 0.85, $P$ < 0.001).

To analyse if similar mutational trends were found outside the capsule operon, we analysed all mutational events in the 164 end-point clones that were sequenced. Non-synonymous point mutations (N) were common and their frequency was constant across strains and environments (36% on average) (Fig. 4B). The ratio of non-synonymous over synonymous changes (S), N/S, revealed an excess of the latter, suggesting that genomes of Kva 342 and Kpn NTUH are under purifying selection (Table S5). We observed that the second most common mutational events in Kva 342 were IS insertions (16%), but gene deletions were very frequent in Kpn NTUH (28%). Strain BJ1 mutated primarily by small base-pair deletions (42%). Further analyses revealed that the frequency of each mutation type depended on both strain ($\chi^2$, statistic = 129.2, df = 18, $P$ < 0.001) and environment, and their interaction ($\chi^2$, statistic = 56.2, df = 36, $P$ < 0.01). Taken together, the results from the experiments both in well-mixed and in structured environments concord to show that, independently of the spatial structure, *Klebsiella* strains evolve and mutate by different strain-specific mechanisms.

### Evolution of hypermucoidy as a by-product of adaptation outside the host

Previous experiments in *P. fluorescens* showed increased viscosity of cells when these evolved in static environments[36,47]. We thus hypothesised that capsulated populations could become hypermucoviscous, a particularly relevant phenotype in *Klebsiella* strains, as it is associated with increased virulence[48]. Indeed, early on the evolution experiment, we observed that populations descending from capsulated ancestors developed the hypermucoid phenotype (HMP) in liquid (Fig. S6A). Such phenomenon was exclusively observed in ASM and LB, the two media with the highest carrying capacity, including in Kva 342, a strain which does not code for the *rmp* locus, that increases capsule expression and is known to cause HMP and hypervirulence[49]. To further test whether HMP could evolve in other environments, we plated all end-point populations to perform the string-test, a hallmark of HMP (Fig. S6B). Our results revealed that, when tested on agar, capsulated populations evolving in all environments, including soil, displayed HMP. In more than half of the populations of environmental strain Kva 342, HMP emerged de novo. However, we also observed that 12 out of the 54 populations derived from the string-test positive ancestral strains BJ1 and NTUH-K2044, no longer displayed this phenotype (Fig. S6B). These results were largely in agreement with HMP quantification by slow centrifugation. Whereas populations of Kpn NTUH, only increased HMP in ASM (Fig. 5B), almost all populations from strain Kva 342 and Kpn BJ1 increased HMP, irrespective of the environment (Fig. 5B) (Two-way ANOVA, Strain ($F$ = 63.1, df = 1) $P$ < 0.001, Environment ($F$ = 1.3, df = 4), $P$ = 0.27). Overall, our results suggest that evolution in structured environments selects for hypermucoviscosity, independently of the presence of the *rmp* locus.

We then investigated the genetic basis of the de novo emergence of HMP. Among the 13 Kva 342 clones sequenced that were capsulated and string-test positive, ten (77%) had single nucleotide polymorphisms in the active sites of the ATPase activity domain of *wzc*, the tyrosine kinase involved in capsule production (Table S6 and Fig. 5C).

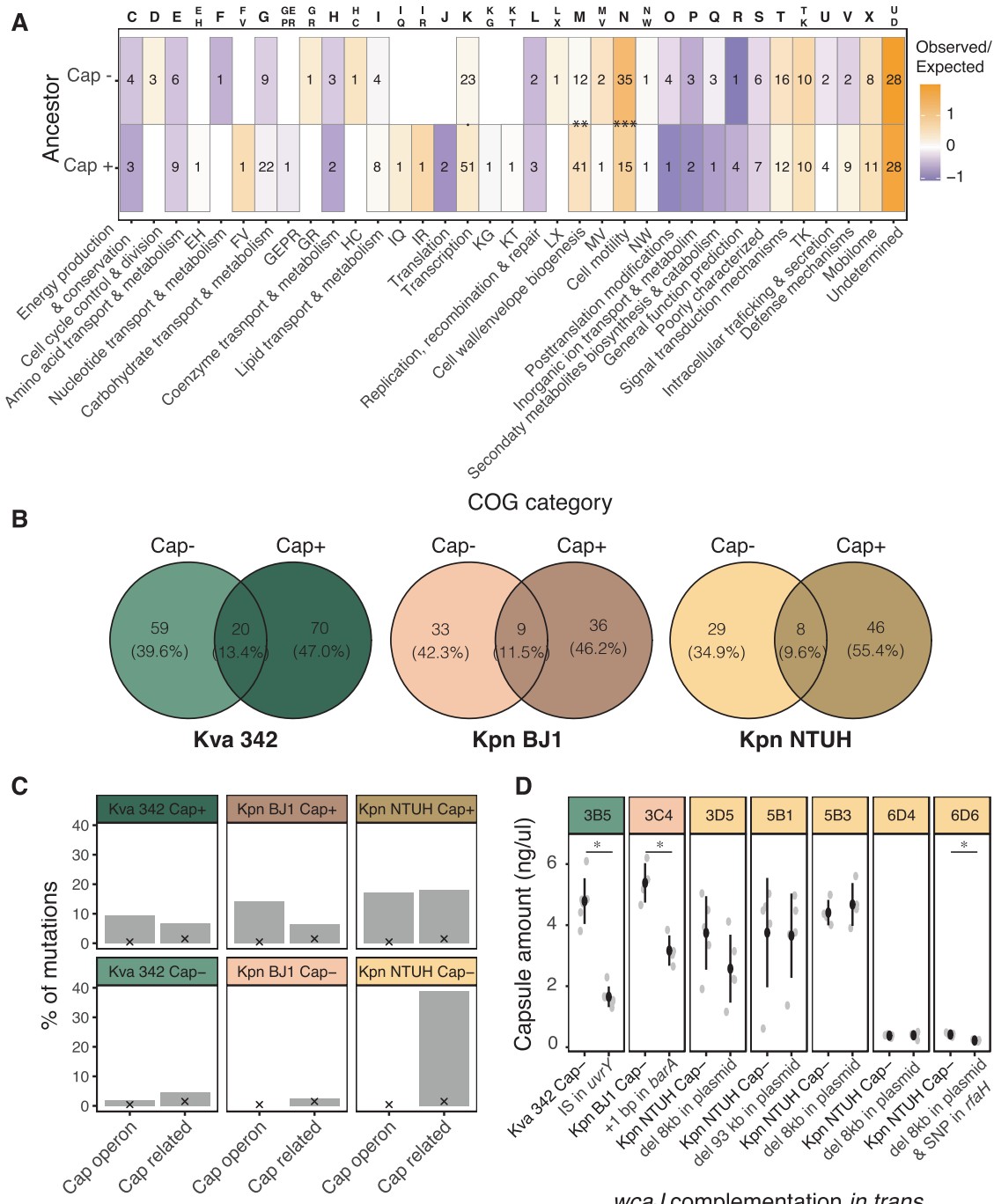

**Fig. 2 | Comparison of mutations across ancestral genotypes. A** Ratio of observed over expected number of mutations in genes annotated to each of the clusters of orthologous groups (COG category). Numbers represent absolute number of mutations. Mutations in intergenic regions and deletions spanning more than one ORF were eliminated from this analysis. Fisher's test. * $P = 0.05$, ** $P < 0.01$; *** $P < 0.001$. **B** Venn diagram illustrating the number of genes mutated in clones from each ancestral genotype and across strains. **C** Percentage of mutations found in the capsule operon or in capsule-related genes. 'X' indicates the percentage of mutations expected under a null assumption of random distribution of mutations across the reference genome of Kpn NTUH[42,43] (see Methods). **D** Capsule quantification of non-capsulated ancestral and evolved clones complemented with *wcaJ* in trans. Each dot corresponds to an independent experiment. 'del' stands for deletion. Clones 3B5, 3C4 and 3D5 evolved from different ancestors in LB, clones 5B1 and 5B3 evolved in M02 and 6D4 and 6D5 in AUM. Error bars indicate the standard deviation from the mean. Two-sided Wilcoxon test, * $P < 0.05$. Source data are provided as a Source Data file.

Introduction of the evolved *wzc* alleles in the ancestral Kva 342 resulted in compact, elastic and string-test positive colonies. These clones had an increased HMP relative to the ancestor (Fig. 5D) but did not produce more capsule (Fig. S7A). Similarly, evolved capsulated clones derived from Kpn BJ1 and Kpn NTUH with mutations in the *wzc* gene exhibited an increased HMP (Fig. S7BC). The convergent evolution of the HMP in structured environments is intriguing because it is a very

costly (~30%) phenotype as measured by growth rate analysis in well-mixed cultures[32] (Fig. S8). This suggests that the large trade-offs between growth rate and HMP can be easily overcome in structured environments. Taken together, our results revealed that hypermucoviscosity can rapidly evolve by point mutations as a by-product of adaptation outside the host and results in strong advantage for the cells in structured environments.

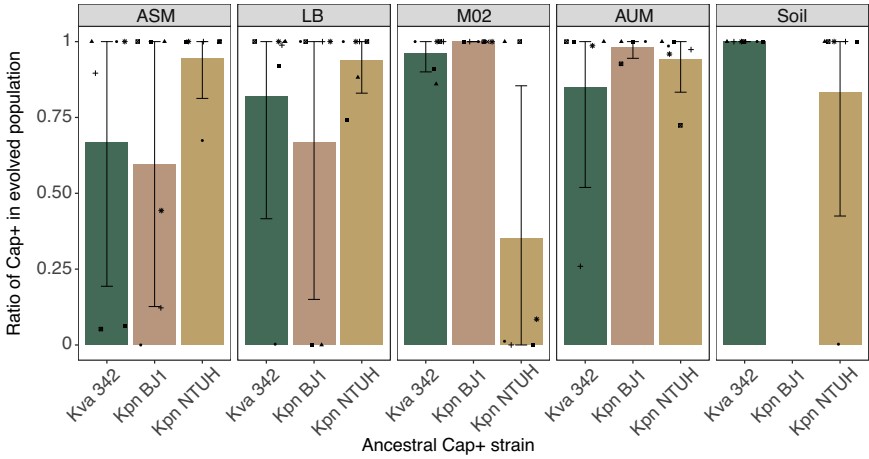

**Fig. 3 | Proportion of capsulated clones in the population at the end of the evolution experiment.** Each dot reflects an independently evolving population. The bar represents the mean of each genotype x environment (up to $N = 6$). Error bars indicate a 95% interval of confidence. Source data are provided as a Source Data file.

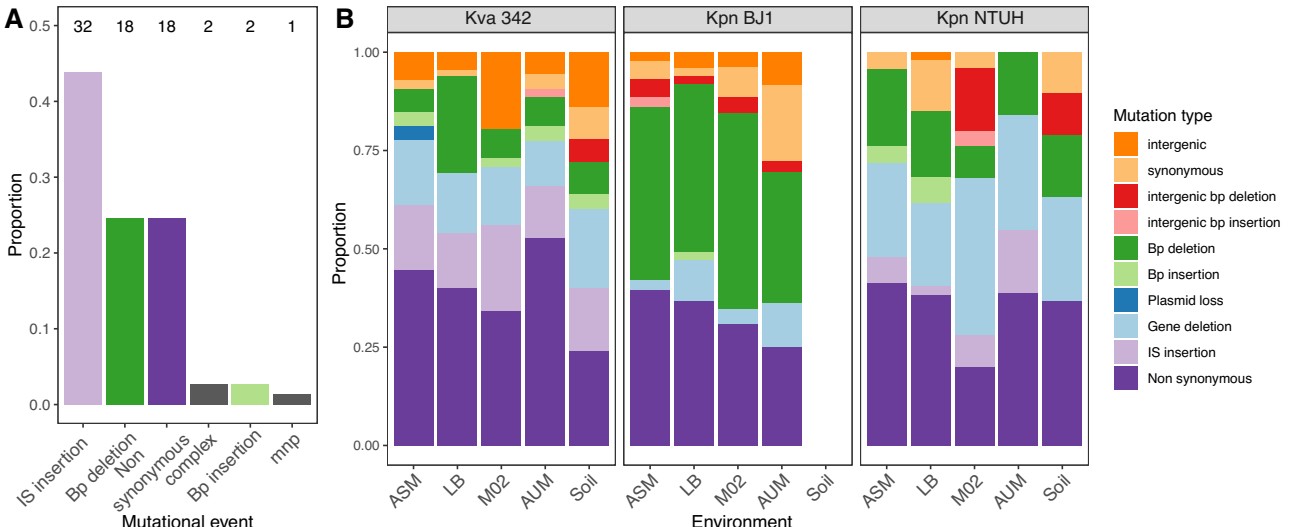

**Fig. 4 | Mechanisms of mutations in evolution experiments of *Klebsiella*. A**. Type of mutations observed in the capsule operon of 73 independent non-capsulated clones descending from capsulated ancestors after 20 generations (end-point) in the short evolution experiment in well-mixed environments. The number on top of the bars indicates how many clones per category. Mnp multiple nucleotide polymorphisms, complex combination of two mechanisms. bp base pair. **B** Type of mutations observed in the 164 clones of the long evolution experiment after ~675 generations (end-point). Source data are provided as a Source Data file.

## Convergent evolution in *ramA* regulon results in increased sensitivity to antibiotics

Our findings that HMP can easily emerge in environments without biotic pressure could be worrisome, particularly because of the recent convergence of hypervirulent and multidrug-resistant clones[50,51]. However, we also observed that the same evolutionary conditions that were selected for HMP also resulted in the accumulation of mutations in the *ramA/romA/ramR* locus, and its regulon (47 mutations in 42 clones of Kva 342) (Table S7 and Fig. 6A). These loci are associated to lipid A biosynthesis, outer membrane stability[52] and resistance to antibiotics[53]. Accordingly, evolved clones with mutations in the locus, had increased sensitivity to ciprofloxacin, tetracycline and chloramphenicol (Fig. S9). Introduction of the evolved alleles in the ancestral background and reversion of one evolved allele to its ancestral sequence confirmed that these mutations were enough to increase the cell sensitivity to antibiotics (Fig. 6B). Such increase is independent of the ancestral background (capsulated or non-capsulated) and is not due to differences in growth rate (Fig. S10).

In ten clones (6%), *wzc* mutations leading to HMP co-exist with those in the *ramA* regulon. However, binomial tests revealed that there is no dependent coevolution between *wzc* mutations and *ramA* ($P > 0.05$ for deviation from the 5% expected under a null assumption that these mutations do not co-evolve). Reversal of *wzc*, *ramA* or both mutations in an evolved clone or insertion of both mutations in the ancestral genotype revealed that, despite an increased HMP, clones with mutations in *ramA* were still more sensitive to antibiotics (Fig. 6C).

Taken together, de novo HMP per se may not be of concern as the same abiotic conditions that selected for it also favour mutations resulting in increased sensitivity to antibiotics.

## Discussion

The *Klebsiella* capsule increases cellular survival under stress or biotic aggressions from other bacteria[17] or the host immune system[54-56]. Yet, there is a lack of studies comprehensively addressing how a major virulence factor such as the capsule, which is widespread across clades,

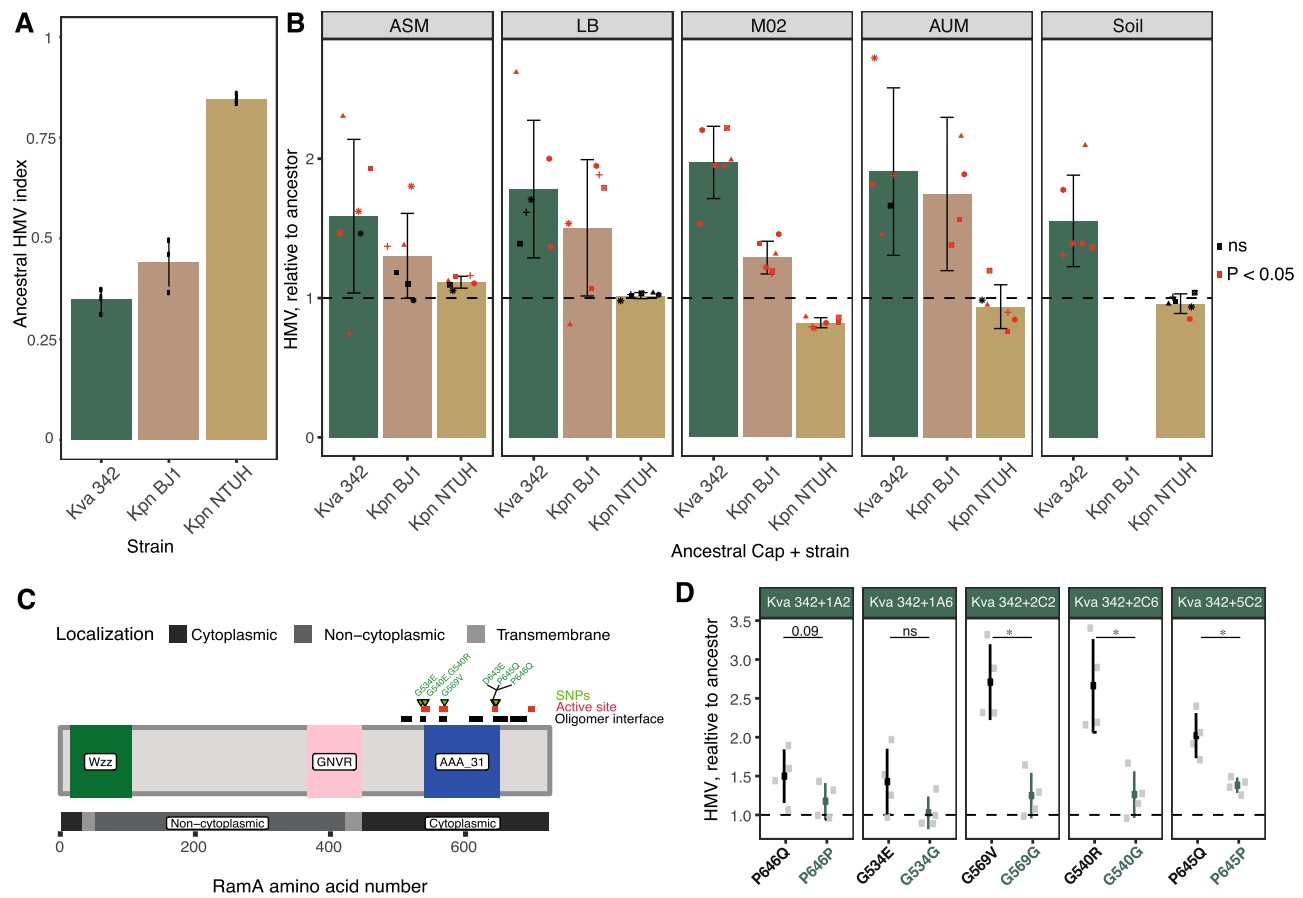

**Fig. 5 | Hypermucoid phenotype of ancestral and evolved clones of *Klebsiella*.**
**A** HMP expressed as hypermucoviscosity (HMV) index of the ancestral clones. The bar represents the mean of three independent biological replicates (individual points) and error bars indicate a 95% interval of confidence.
**B** Hypermucoid phenotype of independent clones relative to its ancestor. Values over one reflect increased HMP. Bars represent the average HMP within treatment and error bars indicate standard deviation from the mean (Treatment statistics provided in Table S1). Each dot reflects the average of at least three independent biological replicates of each independently evolving population. Individual error bars for each dot are not presented for clarity purposes. Red dots highlight clones which are significantly different from the ancestor, One-Sample *t*-test, a difference of 1. ns not significant. Error bars indicate the standard deviation from the mean. **C** Diagram of *wzc* gene, with the predicted functional domains (Wzz, GNVR and AAA_31) from UniProt and cellular localisation of each domain, active site and oligomer interface predicted by InterPro. Figure was constructed with the package *drawProteins* in Bioconductor. **D** HMP is expressed as the HMV index relative to ancestral Kva 324. Five evolved alleles were introduced in the ancestor. After each double recombination event, a clone with an evolved allele (black) and a clone with an ancestral allele (dark green) were kept for comparison. Error bars indicate the standard deviation from the mean. Statistics were calculated using two-sided paired *t*-tests. *$P < 0.05$. Source data are provided as a Source Data file.

affects adaptation to novel environments at the microevolutionary level. Here, we show that the mechanisms of adaptation to novel environments, as well as the associated genetic and phenotypic changes and traits measured in our study, depend on the presence of the capsule (Fig. 1). Ancestral capsulated populations adapted by changes in the capsule operon primarily by increasing hypermucoviscosity, without increasing overall capsule production (Fig. 7). This might suggest that capsule production is already at its maximum level. However, random mutagenesis studies using several strains, including Kpn NTUH, have shown that several mutations can still increase capsule production[42,43]. We also observed the frequent emergence of non-capsulated clones in ancestral capsulated populations, but these rarely fixed and were most often outcompeted by capsulated clones (Fig. 7). This was so in most of the tested environments and contrasts with our experiments in well-mixed environments in which most strains rapidly inactivate the capsule due to a large fitness advantage of non-capsulated clones, in nutrient-rich environments (namely LB and ASM)[32]. Our results are in agreement with a previous study in which *Pseudomonas fluorescens* cells were exposed to several rounds of alternating shaking and static growth. *P. fluorescens* adapted by

evolving a bet-hedging strategy in which a bistable switch controlled the production of a colanic acid capsule[36,57], suggesting that capsules are advantageous in spatially-structured environments but disadvantageous in certain well-mixed contexts. Thus, experiments in two different species, reveal that capsules can be inherently advantageous in the absence of stress or biotic pressure, and that they may have conserved capsule functions across species.

The rapid de novo emergence of hypermucoviscosity, a hallmark of hypervirulent *K. pneumoniae* lineages[38,39,58], in an environmental strain can be achieved by one single nucleotide polymorphism in *wzc*, the tyrosine kinase of the capsule operon. Wzc interacts with the outer membrane transporter, Wza, and controls chain length and polymerisation by cycling between phosphorylated monomers and dephosphorylated octamers[59,60]. However, most mutations do not affect phosphorylated resides, but the ATPase domain found on the cytoplasmic side, suggesting that it is not the polymerisation that is being altered but rather the translocation to the periplasm. Indeed, in clones with mutations in *wzc* and increased HMP, we did not observe higher capsule production as expected if polymerisation were increased, in agreement with studies showing that the HMP is not

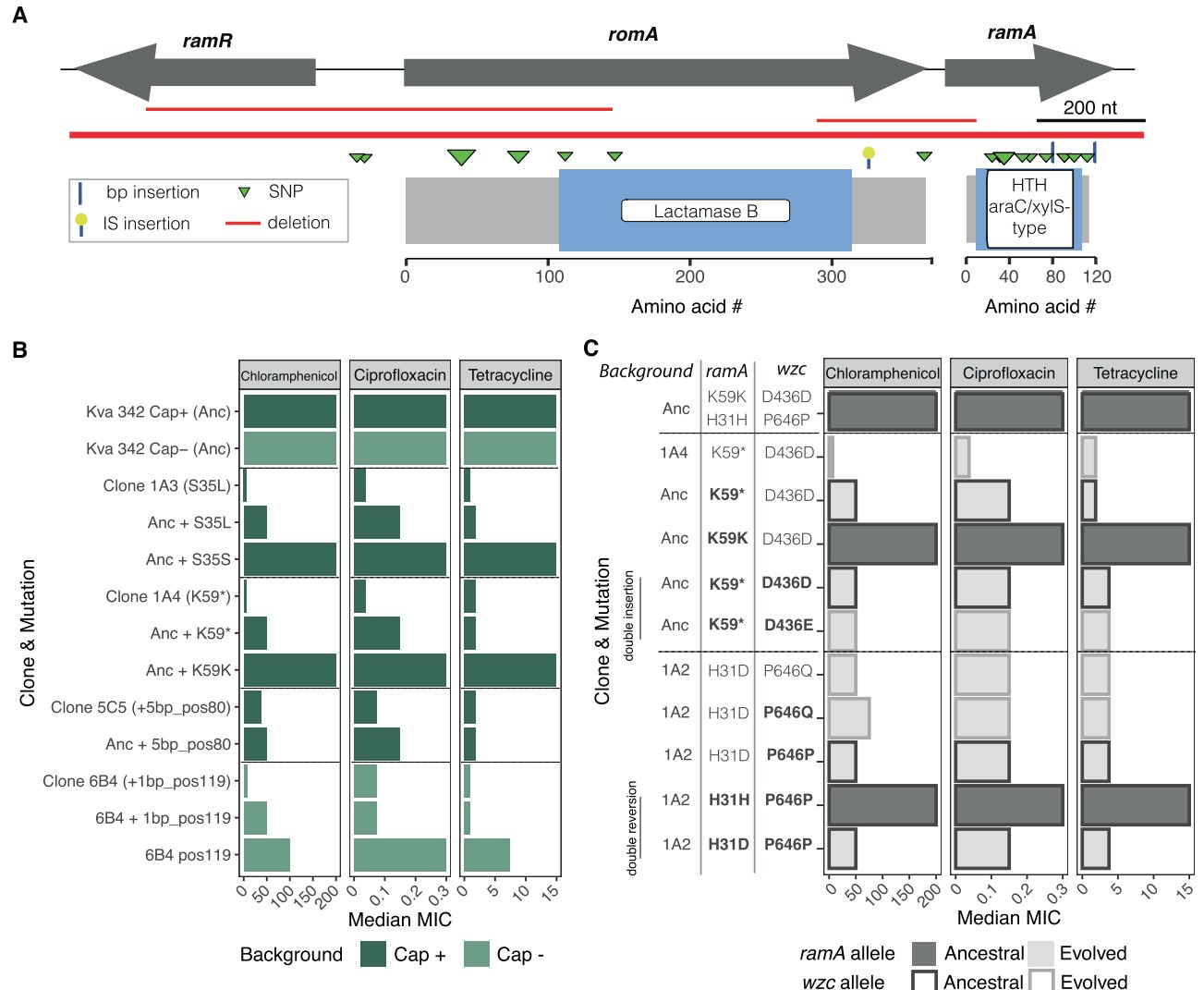

**Fig. 6 | Mutations in *ramA* reduce MIC of several antibiotics. A** Genetic diagram indicating the location of mutations in the *ramA/romA/ramR* locus. The lower panel represents the length and the predicted functional domains of each protein. The position of the triangles (and lines) indicates mutations (and deletions) and their size indicates the number of times the same change occurred independently. **B** Median MIC of several *ramA* mutants and their respective wild-type controls resulting from the same double recombination event. 'pos' refers to the nucleotide position of the mutation in the gene. **C** MIC of capsulated double mutants. The latter were generated either by inserting in the ancestral background first the *ramA* and then the *wzc* allele from the evolved clone 1A4, or by reverting the *wzc* allele first and then the *ramA* allele to the ancestral one in the evolved clone 1A2. The alleles that were modified in each strain are highlighted in bold. MIC is expressed as µg/mL. The median of three independent experiments is represented. Source data are provided as a Source Data file.

always a result of increased capsule production (Fig. S6A). This is also in line with recent research showing that some wzc mutations in trans lead to HMV[61] and it further underlines the complex regulatory pathway leading to the hypermucoviscous phenotype[43,44]. It will be important to test whether the mechanisms of adaptation and the advantage of the capsule are similar in more complex environments, for instance, in mixed bacterial communities. We would expect similar trends, namely, the emergence of HMV in capsulated populations, as the adaptive response is very conserved across the diverse panel of environments tested. Further, it has been shown that the capsule provides competitive advantages during direct competition[16,17] and the costs associated with some traits, like HMV, could be offset by the benefits it provides in a mixed community. Also, similar *wzc* hypermucoid mutants isolated in vivo were shown to mediate phagocytosis resistance and increase dissemination in murine models[62]. This shows that the role of the capsule as a virulence factor may be a result of non-biotic selective pressures outside the host. Moreover, the HMP could hamper predation[63] of *Klebsiella* by amoeba[64] or bacteria[65], as

evidenced by the yeast *Cryptococcus neoformans*, which upon exposure to amoeba predation, developed resistance by increasing capsule size[66]. Additionally, coevolution of *E. coli* with macrophages[67] or with predatory bacteria[68] both result in increased mucoidy, showing that such phenotype is also beneficial outside the host.

Similar mutations in the gene *wzc* were also described in another nosocomial pathogen, *Acinetobacter baumannii* and also rendered clones hypermucoviscous[69] suggesting that HMP could easily emerge in other capsulated ESKAPE pathogens and potentially facilitate the emergence of hypervirulent multidrug resistant lineages. Yet, most epidemiological studies show that transmission of hypervirulence or drug resistance relies on mobile genetic elements and not on mutations[51]. The three following scenarios have been reported in *Klebsiella*: (i) acquisition of multidrug-resistant plasmids by hypervirulent lineages[70], (ii) uptake of virulence plasmids by multidrug resistance lineages[71] or, most worrisome, (iii) convergence of both virulence and antibiotic resistance genes in one plasmid[72]. Here, we pinpoint a possible fourth scenario, whereby MDR strains

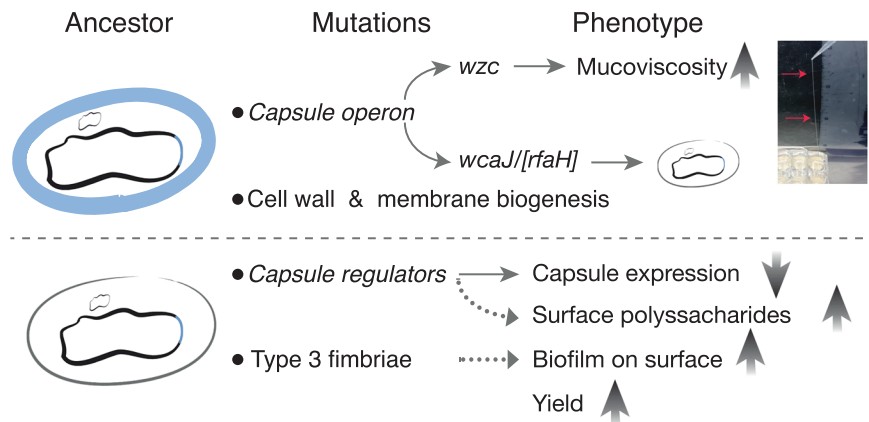

**Fig. 7 | Model of *Klebsiella* adaptation to structured environments.** Capsulated populations adapt by mutations in the capsule operon, which commonly result in increased hypermucoviscosity, and to a lesser extent, capsule inactivation. Additional mutations were found in genes associated with the cell wall and membrane biogenesis. The picture illustrates a positive string-test, a hallmark of HMP identification, performed in liquid. A long bacterial string stemming from the well to the pipette tip can be seen (red arrows). Non-capsulated ancestors primarily evolve (i) by accumulating mutations in capsule regulators that ultimately reduce its expression when they are reacquired by recombination and (ii) mutations in type 3 fimbriae. These populations exhibit increased populations yields and higher production of polysaccharides, which increase biofilm formation. Dotted arrows represent inferred genotype-to-phenotype links.

could easily evolve HMP by mutations in *wzc*. This could have been overlooked in genomic analyses which focus on the prevalence of certain virulence genes, most notably *rmpA, rmpA2*, the *iro* siderophore and the aerobactin *iuc*, but do not analyse core chromosomal determinants of HMV.

Our long evolution experiment also provides a unique opportunity to understand how the absence of a capsule influences the first steps of adaptation to novel environments, which despite the frequent reports of the emergence of non-capsulated clones[20–23,32,33,73] has not been sufficiently addressed. On one hand, non-capsulated cells adapt by increasing the production of other extracellular polysaccharides that could remain tethered to the surface by the action of Wzi, a lectin-binding protein that is still functional. This would suggest that there is a positive selection to conserve a capsule-like function even outside the host and further reinforces the notion that the capsule may also play an important role in *Klebsiella* physiology[43,74]. On the other hand, most non-capsulated clones accumulate mutations in the regulatory elements of the capsule, which may limit the accumulation of toxic intermediate products of capsule biosynthesis[75] and reduce the cost of the expression of other genes in the operon. Such reduced capsule expression could also provide an advantage once the capsule is reacquired by horizontal gene transfer. Indeed, the recovery of a functional capsule operon by homologous recombination can result in a capsule swap, that is, the expression of a novel serotype of a different biochemical composition[23,76]. Because such swaps are not random and are more likely to occur between serotypes with more similar chemical composition[23], this suggests the existence of deleterious interactions between some serotypes and other cellular structures like the LPS[77]. Thus, non-capsulated clones first adapt by limiting capsule expression, which may minimise the potentially destabilising effects of a novel capsule being expressed, allowing the cell to accommodate and further adapt.

Our results also reveal that capsulated and non-capsulated populations adopt different adaptive paths. Specifically, evolved non-capsulated bacteria have higher population yields, surface polysaccharide production and biofilm formation, most likely due to increased production of fimbriae, whereas capsulated populations adapt by increased or de novo emergence of HMP (Fig. 7). Ultimately, these mechanistically divergent strategies of adaptation to novel environments converge in that they result in increased population structure and more cellular interactions. This has the potential to maximise social benefits and maintain cooperation, which is favoured when individuals interact preferentially with other cooperating individuals over non-cooperators[78,79]. Such preferential interactions can be achieved by various means, including increased population structure[80], and physical linkage between cooperating cells, by means of exopolysaccharide secretion[37,81]. Thus, both adaptive mechanisms, HMP and increased biofilm formation may contribute to direct the benefits of cooperation towards cooperative cells[82], because it can physically link cooperative cells or exclude non-relatives from joining the group. Ultimately, we can speculate that adaptation by increased HMV or biofilm formation results in increased social interactions and could be analogous to the emergence of multicellular behaviours and benefits.

Overall, given that most capsulated pathogens are facultative and ubiquitously found in nature and that most capsulated species are free-living[15], our results further support the hypothesis that the fitness advantage of the capsule during infection is a by-product of adaptation outside the host[15,18,74]. Here, we gathered evidence that selection in response to non-biotic physical characteristics such as environmental structure alters traits involved in host-pathogen interactions and affects human disease outcomes. The parallels observed between our evolution experiment in structured environments in terms of the divergent evolution of HMP and non-capsulated clones and the in vivo observations in which both non-capsulated and hypercapsulated clones emerge during *Klebsiella* infection[62] highlight the relevance of using simplified models to understand general adaptive processes. To conclude, the similarities in the genotypic and phenotypic patterns with other species set the capsule as a major determinant of species microevolution, which may affect the many capsulated facultative pathogens, including all other ESKAPE microorganisms.

## Methods

### Bacterial strains and growth conditions

Three different strains from the *Klebsiella pneumoniae* species complex were used in this study (Table 1): one environmental strain, isolated from maize in the USA, *K. variicola* 342 (Kva 342, serotype K30)[28], *K. pneumoniae* BJ1 from clonal group 380 (Kpn BJ1, serotype K2) isolated in France from a liver abscess[83] and the hypervirulent *K. pneumoniae* NTUH-K2044 (Kpn NTUH, serotype K1) from clonal group 23 isolated in Taiwan from a liver abscess[26]. Kanamycin (50 μg/mL), trimethoprim (30 μg/mL) or streptomycin (100 μg/mL for *E. coli* and 200 μg/mL for *Klebsiella)* were used when applicable. All primer sequences are provided in Table S8 and were purchased at Eurofins Genomics Europe.

## Mutant construction

**ΔwcaJ mutants.** (i) ΔwcaJ mutants. Isogenic capsule mutants were constructed by an in-frame deletion of *wcaJ* by allelic exchange, as previously described[32]. WcaJ is the first enzyme of the capsule bio-synthesis pathway and has been shown to be the primary target of mutations leading to capsule inactivation in genomic datasets[23] and in lab-evolved non-capsulated clones[33]. The absence of off-target muta-tions was verified by Illumina sequencing by direct comparison to their respective reference genomes using *breseq* (0.26.1[84]) with default parameters. (ii) Insertion and reversion of SNPs. Ancestral or evolved alleles were amplified by PCR using the Q5 High-Fidelity 2X Master Mix and cloned into linearised pKNG101 with GeneArt™ Gibson Assembly HiFi Master Mix (Invitrogen). The mix was transformed into competent *E. coli* DH5α pir strain, and again into *E. coli* MFD λ-pir strain, used as a donor strain for conjugation into ancestral or evolved clones. Single cross-over mutants (transconjugants) were selected on streptomycin plates and double cross-over mutants were selected on LB without salt and supplemented with 5% sucrose at room temperature. From each double recombination, a mutant and a wild type were isolated. Inser-tion and reversions were verified by Sanger sequencing. (iii) Streptomycin-resistant ancestors. Ten mL of overnight cultures of each ancestral genotype were centrifuged for 15 min at 4000 rpm. The pellet was plated on Streptomycin plates. Individual clones were restreaked, and growth curves were performed in all evolutionary environments to verify that resistance did not result in significant growth defects (Fig. S1BC).

## Evolution experiment

(i) Experimental setup. For each strain, the evolution experiment was initiated from a single colony inoculated in 5 mL of LB and allowed to grow under shaking conditions at 37 °C overnight. Twenty microliters of the diluted (1:100) overnight culture were used to inoculate each of the six independent replicates in the five environments. Each popula-tion was grown in a final volume of 2 mL in independent wells of 24-well microtiter plates (Corning™ 3738, Thermo Fischer) without shaking, as to allow some degree of spatial structure. To limit cross contaminations, every second row was used. We alternated strains to more easily detect cross-contamination. Every 24 h, using filtered tips, 20 μL of each culture was propagated into 1980 μL of fresh media and grown at 37 °C under static conditions in a humid atmosphere to limit evaporation. The experiment went on for 102 days, accounting for *ca* 675 generations. Although each growth media had different carrying capacities, i.e. the maximum population size an environment can sustain, all cultures reached bacterial saturation in the late stationary phase, ensuring that the different populations underwent a similar number of generations across media. Independently evolving capsu-lated populations were plated 28 times, every two days during the first ten days (i.e. days 2, 4, 6, 8 and 10) and every 4 days until the end of the experiment, to follow the dynamics of capsule inactivation.

(ii) Environment description. AUM (artificial urine medium) and ASM (artificial sputum medium) were prepared as described previously[29,30]. AUM is mainly composed of 1% urea and 0.1% peptone with trace amounts of lactic acid, uric acid, creatinine and peptone. ASM is composed of 0.5% mucin, 0.4% DNA, 0.5% egg yolk and 0.2% amino acids. LB is composed of 1% tryptone, 1% NaCl and 0.5% yeast extract. M02 corresponds to minimal M63B1 supplemented with 0.2% of glucose as a sole carbon source. Soil medium was prepared by stirring overnight 10 g/L of commercial potting soil in double-distilled H$_2$O. Prior to autoclaving, the mixture was filtered through cotton to eliminate excess debris. Different environments have different carry-ing capacities ranging from ~ $1 \times 10^9$ CFU/mL in LB and ASM to ~$5 \times 10^6$ CFU/mL in soil, as shown in Fig. S2. (iii) Glycerol storage. Glycerol stocks were prepared for each population at days 7, 15, 30, 45, 75 and 102, corresponding roughly to 50, 100, 200, 300, 500 and 675 gen-erations. (iv) Cross-contamination detection. During the evolution

experiment, cross-contamination tests were performed by PCR using serotype-specific primers based on *wza* capsular gene (Table S8). PCRs were done using the Thermo Scientific™ Phusion Flash High-Fidelity PCR Master Mix and performed every 2 days on two random wells per environment for the first 30 days, then once per week on 20 random wells. Randomisation was carried out by the *sample* function in R. After the evolution experiment, all glycerol stocks were checked for cross contaminations using carbon sources or antibiotics. For carbon sour-ces tests, minimal M63B1 agar was used, supplemented with 0.02% of yeast extract and 0.5% of carbon source. Specifically, we grew popu-lations in minimal media supplemented by tricarbacillic acid (TCBA) and dulcitol to test for the presence of Kva 342 or Kpn NTUH-K2044, respectively. Kpn BJ1 was discriminated by its ability to grow on chloramphenicol at a final concentration of 50 μg/mL, and growth arrest in L-tartaric acid. Out of the 168 populations, we detected con-taminations in four. These populations were excluded from all further analyses.

## Fitness of evolved populations

Competitions between end-point populations and their respective streptomycin-resistant ancestor were performed in three biological replicates, at least. Populations were inoculated directly from the freezer stocks into 4 mL of LB and allowed to grow overnight. The latter were then mixed in a 1:1 proportion with cultures from the respective streptomycin-resistant ancestor grown from an indepen-dent colony. The co-culture was then diluted 1/100 in and 20 μL were inoculated into 1980 μL in the relevant environment. A sample was taken and used for serial dilution and plated on LB (total CFU) and on streptomycin plates to select for the ancestor ($T_0$). After 24 h of competition ($T_{24}$), each mixture was serially diluted and plated on LB and LB supplemented with streptomycin. The fitness of evolved populations was calculated by dividing the proportion of evolved populations at $T_{24}$ over $T_0$.

## Genome analyses

(i) Whole genome sequencing and variant analyses. On day 102, a single randomly-chosen clone from each population was isolated for whole genome sequencing ($N = 164$). This strategy was preferred to whole population sequencing due to the inherent bias of the DNA extraction protocol that would amplify non-capsulated or non-viscous genomes, that would not be representative of clone fre-quencies in the population or may totally mask some mutations of interest. DNA was extracted from pelleted cells grown overnight in LB supplemented with 0.7 mM EDTA with the guanidium thiocya-nate method[85], with modifications. RNAse A treatment (37 °C, 30 min) was performed before DNA precipitation. Each clone was sequenced by Illumina with 150pb paired-end reads, yielding at least 1 Gb of data per clone by Novogene (UK). Each evolved clone was compared to the ancestral sequence using *breseq* (0.26.1[84]) with default parameters. The SNPs identified by *breseq* were further confirmed using *snippy* (https://github.com/tseemann/snippy) with default parameters. When generating appropriate mutants, specific PCR and Sanger sequencing verified that these mutations were present in the evolved clones. Plasmid loss was determined by the analyses of the coverage of the reads on the reference (ancestral) genome as provided by *breseq*. (ii) Pangenome. We used PaNaCoTa[86] to infer the pangenome of the three strains with the connected-component clustering algorithm of MMseq2[87] with pairwise bidirectional coverage >0.8 and sequence identity >0.8. The pangenome comprised 6455 different gene families. If the gene was annotated by *prokka*[88], the nomenclature was kept, otherwise, a family gene number was assigned. (iii) Assignation of COG cate-gories. The COG database was downloaded via NCBI's FTP site (ftp://ftp.ncbi.nlm.nih.gov/pub/COG/COG2014/data/, March 2021). We used the provided python script merger.py to generate a COG

database. BLASTP (2.7.1+) was performed between the three proteomes against the COG database with the following options -evalue 1e-5 -best_hit_overhang 0.1 -best_hit_score_edge 0.05. For each protein, the hit with the highest bitscore and percentage of identity was retained, and its COG category was noted. Intergenic mutations and gene deletions spanning more than one gene were not taken into account in this analysis. (iv) Capsule-related genes. We compiled a reference dataset of proteins that affected capsule production and HMP based on two previous studies in which in-depth mutagenesis was performed and capsule production assessed[42,43]. We downloaded the protein sequences of genes shown to affect the capsule and performed a BLASTP (default parameters) against all proteins in which mutations were found in the evolved clones. In the case of intergenic mutations, we included the proteins that resulted directly from the upstream or downstream genes, as we had previously observed mutations in the *ops* element, upstream of the capsule length regulator *wzi*, known to regulate the capsule. Proteins that shared more than 80% identity with proteins from the reference dataset were considered as capsule related or belonging to the capsule operon. From the 5,248,520 bp of the Kpn NTUH genome, 103,320 bp (-1.9%) belong to coding sequences known to synthesise or influence capsule production[42,43], including the 24,985 bp from the K1 capsule operon. (v) Detection of IS elements. Genomes were mined for IS elements using ISfinder database[25]. To discard redundancy, for each IS element, we verified 10 bp up and downstream for the identification of another element and selected that with the highest bitscore. We then discarded elements with an e-value higher than 10e$^{-20}$ or less than 750 bp in length.

### *wcaJ* complementation
To restore capsule production in evolved clones derived from non-capsulated ancestors, the corresponding *wcaJ* gene was cloned into the expression vector pUCPT28 (Trimethoprim resistant, TmpR) by Gibson assembly. Briefly, the pUCPT28 vector was first linearised by PCR using the Q5 High-Fidelity 2X Master Mix (New England Biolabs) (primers in Table S8) and digested by DpnI restriction enzyme (New England Biolabs) for 30 min at 37 °C. *wcaJ* was amplified by PCR as above, cloned into pUCPT28 using the GeneArt Gibson Assembly HiFi Master Mix (Thermo Fischer) and transformed into competent *E. coli* DH5α-pir strain. TmpR colonies were isolated and the presence of the insert was checked by PCR. Empty pUCPT28 (control) and pUCPT28 with *wcaJ* were transformed into the corresponding competent evolved clones and, when relevant, verified for capsule production.

### Trait quantification in each environment
To initiate the different measurements, each population was grown overnight and 20 µL of each culture was inoculated into 1980 µL of the relevant growth media in 24-well microtiter plates and allowed to grow for 24 h without shaking at 37 °C. Populations evolving in poor media (M02, AUM and soil) were diluted 1:100 and allowed to grow for another 24-h extra prior to the experiment, as we noticed that preconditioning was important for reproducibility. (i) Population yield. Each well was homogenised by vigorous pipetting and then serially diluted in fresh LB and plated to count CFU after 24 h of growth. For most capsulated populations in LB and ASM, due to the extreme hypermucoviscous phenotype (HMP, Fig. 7), CFU could not be accurately assessed as the populations cannot be resuspended and homogenised. Thus, the serial dilution process was biased and distorted because either a randomly large or a randomly small proportion of the population would be transferred due to the abovementioned HMP. (ii) Biofilm production. After 24 h of growth in the evolutionary environment, unbound cells were removed by washing once in distilled water. To stain biofilms, 2100 µL of 1% crystal violet was added to each well for 20 min. The crystal violet was decanted and washed thrice with distilled water. After the plate

was totally dry, the biofilm was solubilized for 10 min in 2300 µL of mix with 80% ethanol and 20% acetone. Two hundred microlitres of each mix was transferred in a well of a 96-well plate. The absorbance of the sample was read at OD$_{590}$. (iii) Capsule extraction and quantification. The bacterial capsule was extracted as described in ref. 89 and quantified by using the uronic acid method[90]. Briefly, 500 µL of each population (1500 µL for AUM) from each well were centrifuged and the supernatant was discarded as to specifically measure cell-bound surface polysaccharides. The uronic acid concentration in each sample was determined from a standard curve of glucuronic acid. To quantify the capsule production of *wcaJ* complemented strains, empty vectors were transformed into all *wcaJ* strains (evolved and ancestral). To reduce the noise produced by other surface-attached polysaccharides in evolved clones, we calculated the difference between clones transformed with the vector expressing *wcaJ* and the empty vector. To measure surface polysaccharides in non-capsulated mutants, the same protocol was followed.

### Hypermucoviscous index
Each culture was preconditioned in LB for 6–8 h, diluted at 1:200 (final volume of 4 mL) in M02 and allowed to grow overnight under shaking conditions at 37 °C. After vigorous vortexing, 200 µL of each culture was transferred into a 96-well plate and absorbance was read at OD$_{590}$ (OD$_i$). The tubes were then centrifuged for 5 min at 2500×*g* and a second absorbance was read at OD$_{590}$ (OD$_f$). The HMP index was obtained using the ratio $\frac{ODf}{ODi}$.

### MIC calculations
Minimal drug inhibition concentration was calculated in a liquid medium by microdilution as recommended by EUCAST guidelines[91]. Antibiotic stock concentrations were 1000 mg/L or greater and diluted further to working concentrations. Overnight cultures in Mueller–Hinton (MH) liquid for each strain were diluted to reach a final concentration of $5 \times 10^5$ CFU/mL. Diluted cultures and antibiotics at working concentrations were mixed in a 1:1 ratio to a final volume of 100 µL in microtiter plate wells. The MIC was determined as the lowest concentration of the antibiotic that completely inhibits visible growth as judged by the naked eye after 24 h of incubation at 37 °C in static conditions. Each measurement was performed thrice and CFUs controlled, as it can have a major impact on MIC determination.

### Data analysis
All the data analyses were performed with R version 3.5.3 and Rstudio version 1.2. Statistical tests were performed with the base package stats. We used multifactorial ANOVA and a bidirectional stepwise regression model to test whether genotype, capsule or environment influenced evolutionary changes in phenotypic traits. Student-*t*-tests were used to test for differences between independent populations, clones or within-treatment averages with respect to their ancestor or between evolved and ancestral alleles. Fisher's exact test was used to compare whether there were significant differences in the amount of mutations per COG category in capsulated and non-capsulated clones. Binomial tests were used to check whether mutations in certain genes/loci were more common than expected by chance. For data frame manipulations, we also used dplyr v0.8.3 along with the tidyverse packages. All experiments were performed at least three independent times.

### Reporting summary
Further information on research design is available in the Nature Research Reporting Summary linked to this article.

## Data availability
All data generated in this study have been deposited in the public repository Figshare at: https://figshare.com/articles/dataset/RawData_

Nuccietal_2022/19597195 (ref. 92). Raw reads for this project can be accessed in the European Nucleotide Archive (ENA), project number PRJEB54810. Source data are provided with this paper.

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

## Acknowledgements

We thank Matthieu Haudiquet and Jorge M. de Sousa for helpful discussions. We are grateful to Gan Yunn Hwen for the gift of the pUCP28T plasmid used for the complementation of *wcaJ*. This work was funded by an ANR JCJC (Agence national de recherche) grant [ANR 18 CE12 0001 01 ENCAPSULATION] awarded to O.R. The laboratory is funded by a Laboratoire d'Excellence 'Integrative Biology of Emerging Infectious Diseases' (grant ANR-10-LABX-62-IBEID) and the FRM [EQU201903007835]. The funders had no role in study design, data collection and interpretation, or the decision to submit the work for publication.

## Author contributions

O.R. conceived and designed the details of the study with input from E.P.C.R. O.R. and A.N. performed the experiments and the statistical analyses. O.R. performed the bioinformatics work, analyzed the data and wrote the first draft of the manuscript. O.R. and E.P.C.R. secured funding, provided the resources and materials necessary for this study and revised the manuscript. All authors approved the final version of the manuscript.

## Competing interests

The authors declare no competing interests.
