## [Peer Review File · Nature Communications]

Adaptation to novel spatially-structured environments is driven by the capsule and alters virulence-associated traitsReviewers' Comments:

Reviewer #1:

Remarks to the Author:

Nucci et al carry out an extensive series of experimental evolution studies using WT and non-capsulated strains of *Klebsiella*, with the aim of dissecting the role of capsules in bacterial adaptation. Their overall objective is to examine the idea that the capsule's effects on virulence arise as a byproduct of selection in non-host environments. The authors evolve 3 genotypes (WT and non-capsulated) in 5 different media for almost 700 generations. After evolution, strains are sequenced and tested for CFU, biofilm formation, and mucoidy. In short, they find that the presence of a capsule partly determines evolutionary outcome. While strains with a capsule tend to increase mucoviscosity, a non-trivial fraction also lose capsular expression by a variety of molecular mechanisms in capsular genes (although these mutants don't tend to fix). By contrast, strains without a capsule decrease capsular expression via changes in capsule regulation, presumably to reduce costs or toxicity of a non-functional capsule, while also increasing in yield and biofilm formation.

Although the experiments are well designed and the results are detailed and interesting, the paper lacks any clear hypotheses, aside from the very general notion that the capsule somehow matters for adaptation. The media conditions and strains seem arbitrary, and don't really figure in the analyses, and the effects of spatial structure are assumed rather than tested, even though this aspect is highlighted in the opening paragraph. These and other specific concerns are outlined below:

More major issues:

- 1) Spatial structure is introduced by using non-shaking cultures, in contrast with earlier experiments from the group. I like the idea here, but the effects of spatial heterogeneity are never really examined. Spatial structure could/should facilitate population diversity and possibly coexistence of strains with different strategies (following the long line of papers from Paul Rainey, some of which are referenced), but traits of evolved populations are only measured at the level of the entire population and only a single clone from each population is sequenced. This makes it impossible to assess if e.g. non-capsulated strains, which don't fix, persist with capsulated/hypermucoid strains, or if the mutations that arise in either population affect fitness in isolation or in the context of the rest of the population. By only examining single isolates/populations, the power and interest of introducing spatial structure is largely lost. This makes it difficult to draw clear conclusions about the apparent difference in results between this and their earlier experiments.
- 2) Media conditions are diverse, but there is very little consideration of whether and how results depend on these media options. Do strains evolve differently in environments that "mimic host-related nutritional conditions" (line 81) from those that don't (both in terms of phenotypes or the mutations that arise)?
 - a. The paper notes that populations are diluted in the same way in each population, thereby ensuring an equal number of generations. But it could very well be that overall population sizes differ markedly in different media, and potentially for different genotypes or with respect to capsule (yield is clearly a target of selection). These need to be clarified as population size can strongly influence evolutionary outcomes.
- 3) Phenotypic results are, overall, difficult to discern from either the text or the figures (figure 1 mainly); many of the results are only given in the SOM. Figure 1, and others throughout, focus on relative changes. This makes some sense as a means of normalizing the results across strains and conditions. But it also makes it difficult to evaluate the absolute changes which may vary as a function of the starting phenotype. Figure 4 shows that this can clearly matter; strains that are highly hypermucoid change less than strains that aren't. I'd suggest a similar presentation as Figure 4 for the other phenotypes as well.

More minor issues:

- 1) The abstract suggests that the capsule “shapes adaptation to novel niches”. I think this overstates the case. What is tested is that the capsule isn’t only about virulence.
- 2) Perhaps this is a matter of terminology, but I found it confusing to speak of reduced capsular expression in a strain lacking a capsule. It would help to more clearly distinguish the capsule itself from capsular genes.
- 3) There is discussion of “compensating” for the absence of a capsule, but it isn’t clear what they are compensating for. Is this referring to fitness or something else? Some non-capsular strains produce excess polysaccharide. Is there any evidence that this recovers functions once served by the capsule?
- 4) Yield increases in populations without a capsule, but it isn’t clear if the non-capsulated strains have higher yield than their WT parents to begin with. My naïve prediction would be that that capsule is costly, and so yield would start higher in these strains. Further degradation of a non-functional pathway would then be beneficial. The paper notes that these traits evolved by co-evolution. I think this is better referred to as trait correlations and pleiotropy—1 change affecting several phenotypes and not two traits evolving in concert.
- 5) Trait categories in Fig 2A should be spelled out rather than using the COG categories. This should at least be done for the categories where differences are observed.
- 6) I understand the need to distinguish KlebEvo 1 and KlebEvoII internally, but I found the designation unnecessary and distracting in this paper. It’s easier to say “we previously found...”, etc.
- 7) From Line 244 the paper looks at mutations leading to capsular loss. These are interesting data, but the justification is unclear (as is the interpretation of the comparison) if the expectation is that structure matters for phenotype and their causal mutations. The conclusion on line 271 is therefore somewhat vague “Taken together, our results show that, independently of the environmental structure, Klebsiella strains evolve and mutate by different mechanisms”. This would benefit from some elaboration.
- 8) Social arguments/bet-hedging/etc are not well supported with data. There is an assumption that evolved strains have increased population structure, more social interactions, and that this affects cooperation. These are really interesting, and perhaps likely, possibilities. I hope they will be tested in future work with these strains, b

Reviewer #2:

Remarks to the Author:

Dear Authors,

I enjoyed reading your paper and I think it makes an important contribution to our understanding of how Kleb capsule positive and capsule negative variants adapt to novel environments. I have a couple of comments that largely relate to clarity of meaning regarding the text plus additional comments on statistics reporting, sequence analysis and discussion.

1. Grammar and syntax

Throughout the text the “adaptation to” and the “first steps in adaptation” is missing the qualifying words which is “novel environments”? They are not just randomly adapting to nothing and as such I think that you need to be clear throughout the text as to what they adapting to both in the general sense of the paper i.e. novel environments and then with regard to specific contexts i.e. soil, ASM etc...

As an example

Line 27 adaptation primarily occurs.... I think that this should read adaptation to novel environments and line 69, 429

Lines 40-43 This sentence is very long and confusing, it needs to be broken down and/or rephrased

Line 45 how components at the cell surface.... This is very vague, what are you referring to, what are the cell surface components?

End of sentence in line 55 – this is a statement of fact and needs a reference

Lines 56 to 59 starting at Metagenomic analysis... I think that this is a little bit confusing and could do with be better qualified and referenced.

Line 60 change todiverse animals, but it is also found in.....

Line 62 needs a reference

Line 68 change to strongly influences the

Line 103 (HMP)

Line 105 spell out CFU

Line 308 Qualify your argument more clearly.... i.e. It is a very costly phenotype as measured by growth rate analysis....

Line 365 No study has....

Line 367 to 368 This sentence is both confusing and overstated.... You looked a number of phenotypic traits not all phenotypic traits possible... I would change this toHere, we show that the mechanisms of adaptation, as well as the associated genetic and phenotypic changes and traits measured in our study, depend on the presence of capsule.

Line 517 to 519 I found this sentence very confusing, it is not very clear to me what days you were actually plating at ...eg. Every second day during ten days???

Line 544 – did you sequence 180 isolates? I would but the n = X here.

2. Statistics

I would like to see a full breakdown of how each statistical test in the methods not just the package that was used to calculate them. i.e. We used Fisher's Exact to test for X, Y and Z..... Two-way ANOVA for X, Y and Z.... and so on.

For all statistical analysis reported in the results and text please report in full e.g for ANOVA give F statistic, df and p.

I am not familiar with the inclusion of N, K and M prefixes before reporting the Fisher's test result? Also prefixes are missing for the reporting of this stat elsewhere

3. Other comments:

Line 120 Capsule production either remains near ancestral levels or is strongly diminished... could capsule production be at its maximum level of production?

Figure 1. I found this figure very small and hard to read at the current size. Also please change the symbols for the strains as the colour for the two Kpn strains looks the same to me (even if the symbol is a different shape it is important to make them highly distinguishable)

Line 334 Our findings that HMP....without biotic pressure....

This is very interesting but I think it also raises an important point – in natural environments Kleb strains will be embedded in complex microbial communities (other bac, phages etc) and therefore the evolution of costly traits may be constrained if there is competition etc. Whilst experimental evo is very important to hypothesis testing it often uses highly reductionist experimental setups as you outline in this paper and whilst your experimental setup is not an issue (and the use of multiple abiotic environments is great), and does not preclude its acceptance, I think it is important to elaborate on the importance of community context in the evolution of the specific traits you measure (growth/pop size) and if you would anticipate similar results if there was e.g. a background community present in the discussion section.

Sequence analysis – I have used Breseq and found that it generated several false positives and also failed to identify actual mutations when using the default setting. I would recommend complementing your analysis with Samtools or another programme. Also Breseq is primarily useful for indels and SNPs but I see that loss of plasmids were reported in Figure 4 - how were these detected?

Reviewer #3:

Remarks to the Author:

This is an interesting piece of work examining the evolution and adaptation of capsulated bacteria under several different conditions, and attempting to draw general principles that drive selection of certain traits that may or may not impact virulence. Its premise and setup of experiments are innovative and derive a lot from prior work of the same authors, further sharpening their arguments and providing more evidence. Deciding to examine capsule + and null strains in the evolution experiment is a very good decision and indeed the authors find that such background contributes to the evolution of different mutations that seems to be selected for different traits. The selection of traits to be examined is also very sound. However, there are some points that need to be further clarified and addressed.

1. It is unclear what spatially structured environment means. Do the authors mean shaking versus no shaking? Or in fact means specific kinds of media? Why would eg AUM be considered spatially structured versus LB?
2. It is unclear why cfu cannot be measured for HMP strains. How is cfu determined? Is it not by plating? It is not explained in methods. Or are authors referring to Fig 3 on deciding whether it is capsulated or uncapsulated?
3. Why can't HMP be measured for those evolved in LB? Could not the HMP assay be modified for differential centrifugation rather than reading OD of unpelleted vs pelleted bacteria in the original media? These data may be quite interesting to give more context to the evolution of traits.
4. How is increased surface polysaccharides in non capsulated strains measured?
5. Figures 1-3, there are big differences between strains BJ1 and NTUH. The authors did not explain nor discuss about these 2 strains.
6. Figure 5 is fascinating. In the wzc mutated evolved strains, did the authors check NTUH rmpA mutant would then give a more similar phenotype as the other strains showing evolved HMP in all media? Does the absence of rmpA or rcs gives a different ratio?
7. I find figure 6C very hard to understand. Can the authors think of a better way to represent the data? Confusing to know the letters represent clones and mutations in which genes.
8. The inability to reverse phenotypes or effect phenotypes in the wzc and ramA in Figure 6 for

antibiotic susceptibility is a little disturbing. Can the authors examine their evolution expts in different timepoints to see indeed wzc appear first before ramA? Or vice versa? And complement the change singly to test antibiotic susceptibility?

9. It is unclear whether the authors examined growth rates of mutations seen in figure 6 as a reduced growth rates could result in antibiotic susceptibility as measured by MIC, being a more tolerant phenotype.

Reviewer #1 (Remarks to the Author):

Nucci et al carry out an extensive series of experimental evolution studies using WT and non-capsulated strains of *Klebsiella*, with the aim of dissecting the role of capsules in bacterial adaptation. Their overall objective is to examine the idea that the capsule's effects on virulence arise as a byproduct of selection in non-host environments. The authors evolve 3 genotypes (WT and non-capsulated) in 5 different media for almost 700 generations. After evolution, strains are sequenced and tested for CFU, biofilm formation, and mucoidy. In short, they find that the presence of a capsule partly determines evolutionary outcome. While strains with a capsule tend to increase mucoviscosity, a non-trivial fraction also lose capsular expression by a variety of molecular mechanisms in capsular genes (although these mutants don't tend to fix). By contrast, strains without a capsule decrease capsular expression via changes in capsule regulation, presumably to reduce costs or toxicity of a non-functional capsule, while also increasing in yield and biofilm formation.

Although the experiments are well designed and the results are detailed and interesting, the paper lacks any clear hypotheses, aside from the very general notion that the capsule somehow matters for adaptation. The media conditions and strains seem arbitrary, and don't really figure in the analyses, and the effects of spatial structure are assumed rather than tested, even though this aspect is highlighted in the opening paragraph. These and other specific concerns are outlined below:

We thank the reviewer for the comments which have most certainly helped to clarify and focus our manuscript.

We have significantly revised our manuscript to clearly highlight our main hypotheses and motivations of this work:

1. We wanted to test whether the capsule influences evolutionary outcomes during the first steps of adaptation to novel environments, as our previous bioinformatics work strongly suggested.
2. We tested whether the generic process of adaptation could influence traits under selection, such as yield, but also other virulence-associated traits, and whether these were contingent on the presence of the capsule, even in environments in which there was no selective pressure from an immune system. This also derives from our bioinformatics predictions which suggest that capsules could evolve as a by-product of adaptation and not solely due to the advantages within a host.

We have now better justified the choice of the strains and environments in the manuscript and how this was best suited to test our hypotheses.

1. To highlight the general patterns of the adaptive process, rather than the specificity of how one particular strain adapts to one specific environment, we chose three phylogenetically-distant strains and five very diverse environments. More specifically, we chose two *K. pneumoniae* strains from two different clonal groups each with one of the two capsules mostly associated to hypervirulence (namely K1 and K2). To ensure that we were tackling generic evolutionary outcomes common to the genus and not specific to virulent strains, we also chose an environmental *K. variicola* strain.
2. Concerning the evolutionary treatments, we decided to focus on some host-mimicking environments with different carrying capacities and soil, where *Klebsiella* can be readily found as well as two more simple environments (LB and M02), in order to maximize environmental diversity.

3. We performed our experiment in non-shaking environments. Although this cannot fully recreate the complexity of many structured environments, we reasoned that it would be more relevant than shaking cultures. Our goal was not to compare how spatial structure impacts evolution of *Klebsiella*. We insist several times that populations are adapting to novel environments with some degree of spatial structure, as it is important to define the specificities of our evolutionary treatments (see comment 1 from reviewer #2), as opposed to other evolution studies in the field performed in well-mixed environments.

We do however compare the mutational events leading to capsule loss with a previous study in which we evolved strains in a well-mixed culture to show that the mechanisms that we observe here with three strains are the same observed in a larger panel of *Klebsiella* strains (16 different strains). However, direct comparison between the two studies in mechanisms of adaptation to novel environments cannot be performed as the short evolution study in shaking conditions only ran for 3 days (~20 generations) as opposed to the ~700 generations of the study described on this manuscript.

We have also streamlined our text to avoid any reference to well-mixed environments when unnecessary and reduced the references to spatial structure in the first paragraph of the introduction.

More major issues :

- 1) Spatial structure is introduced by using non-shaking cultures, in contrast with earlier experiments from the group. I like the idea here, but the effects of spatial heterogeneity are never really examined.

See our response above.

Spatial structure could/should facilitate population diversity and possibly coexistence of strains with different strategies (following the long line of papers from Paul Rainey, some of which are referenced), but traits of evolved populations are only measured at the level of the entire population and only a single clone from each population is sequenced.

The reviewer raises an important point. We do agree that the emergence of diversity and within population dynamics are central questions in evolutionary biology, but they are not the focus of this manuscript. We believe that further analyses on this matter along the lines of P. Rainey's, V. Cooper's, L. Forney's experiments or I. Gordo's theoretical work, would not bring much novelty on fundamental grounds, as they have done ample research on the topic. Such analyses would also distract from the main point of our manuscript, which is the role of the capsule.

Deep sequencing of a population is feasible. However, genome extraction of the population would be very biased towards the genomes of non-capsulated clones, due to the inherent difficulty of extracting genomes in mucoid or capsulated samples, and would not provide a representative picture of the population.

This makes it impossible to assess if e.g. non-capsulated strains, which don't fix, persist with capsulated/hypermucoid strains, or if the mutations that arise in either population affect fitness in isolation or in the context of the rest of the population.

In Figure S2 (Figure S3 in the revised manuscript), we follow the emergence of non-capsulated strains throughout the experiment, and thus, indirectly show the coexistence between capsulated and non-capsulated genotypes, and how these dynamics change in function of time. In some populations, we do observe the coexistence between non-capsulated and capsulated clones for several hundreds of generations, but we also observe some (few)

selective sweeps of non-capsulated clones including in populations in which hypermucoviscous clones were identified (In Kpn BJ1 ASM population 1, LB population 2 & 3 or in Kva 342 in LB, Population 1). We now comment more on the dynamics of coexistence observed in populations with capsulated and non-capsulated clones in the text.

Concerning fitness in isolation of capsulated and non-capsulated, we have already done some work using a large panel of *Klebsiella* strains (including the three strains used in this study), see Buffet et al, 2021, Proc R Soc B, in shaking cultures, showing that non-capsulated clones are fitter than capsulated clones in nutrient rich environments (ASM and LB). However, this does not seem to be the case in non-shaking environments, as in most populations, the populations remain capsulated, as reported in the results section and Figure 3.

By only examining single isolates/populations, the power and interest of introducing spatial structure is largely lost. This makes it difficult to draw clear conclusions about the apparent difference in results between this and their earlier experiments.

We respectfully disagree. Our goal was to know if the presence or absence of the capsule led to different adaptive pathways when evolving in a structured environment. We show that it does. While we agree that understanding diversity within populations is interesting, we don't think that such analyses would affect the main conclusions of our work, namely that non-capsulated populations increase production of surface polysaccharides and biofilm formation, and tend to accumulate mutation in capsule regulators, whereas capsulated clones tend to evolve hypermucoviscosity by virtue of mutations within the capsule operon (*wzc*).

To streamline our message and avoid further confusion, we have now checked our text and deleted references to well-mixed environments other than in the section of mechanisms of mutation (and discussion).

2) Media conditions are diverse, but there is very little consideration of whether and how results depend on these media options. Do strains evolve differently in environments that “mimic host-related nutritional conditions” (line 81) from those that don't (both in terms of phenotypes or the mutations that arise)?

Our goal was not to compare artificial versus representative environments, but to use media that are representative of the ones it finds in the environment in complement to the standard lab media. We have however performed *post hoc* tests when the effect of the environment was significant and specifically tested for differences between host-associated (ASM and AUM) and other (LB, M02 and soil). We do not observe significant differences between the two groups. We now mention this in the manuscript.

a. The paper notes that populations are diluted in the same way in each population, thereby ensuring an equal number of generations. But it could very well be that overall population sizes differ markedly in different media, and potentially for different genotypes or with respect to capsule (yield is clearly a target of selection). These need to be clarified as population size can strongly influence evolutionary outcomes.

The reviewer is correct. Different media have different carrying capacities which may influence evolutionary outcomes. Absolute values of population yield are now provided in the new supplementary figure (Figure S1) showing the precise carrying capacities for each ancestral genotype in each media. These range from $\sim 10^9$ CFU/mL in ASM and LB to $\sim 5 \times 10^6$ CFU/mL in soil. We now mention this in the methods section. Of note, most mutational events analyzed, notably those inactivating the capsule as well as mutations in the *ramA*

regulon emerge both in the environment with highest carrying capacity (LB) and in the lowest (Soil).

3) Phenotypic results are, overall, difficult to discern from either the text or the figures (figure 1 mainly); many of the results are only given in the SOM. Figure 1, and others throughout, focus on relative changes. This makes some sense as a means of normalizing the results across strains and conditions. But it also makes it difficult to evaluate the absolute changes which may vary as a function of the starting phenotype. Figure 4 shows that this can clearly matter; strains that are highly hypermucooid change less than strains that aren't. I'd suggest a similar presentation as Figure 4 for the other phenotypes as well.

Although we understand the point raised by the reviewer concerning absolute values, we believe normalizing all results is essential for the understanding of the direction and magnitude of evolutionary changes. Further, most differences in absolute values are not due to strain background or trait improvement, but to environment (see figure below in which we show Figure 1 but in absolute values).

Further, absolute values strongly influence hypermucooidy due to the specificities of the test, in which there is a maximum value that can be measured (*ie.* 1), thus limiting trait improvement beyond that. This is not so in biofilm formation, yield or capsule production. In addition, total production of surface polysaccharides should not be compared across strains, as the amount of glucuronic acid present in the polysaccharides from each strain may vary and thus may not necessarily represent more or less production across strains (but this remains significant within strains). Concerning CFU, all three strains have a similar yield across the different media (see above).

We follow the reviewer's suggestion and we now provide a new Supplemental figure (Figure S1) in which we show the absolute values of the ancestors for biofilm formation, surface polysaccharide production and population yield as in Figure 5. We also report all relative values for each population in each environment for each trait, as well as within-treatment means, as in Figure 5.

More minor issues:

1) The abstract suggests that the capsule “shapes adaptation to novel niches”. I think this overstates the case. What is tested is that the capsule isn't only about virulence.

We do not fully agree with the reviewer's claim. Here, we evolved different *Klebsiella* strains in five different environments and primarily test how capsulated and non-capsulated strains adapt to those novel environments and what are the genetic mechanisms underlying adaptation. This is an important point also highlighted by reviewer #2 (see comment 1.a of reviewer #2). Because the capsule is maintained even in the absence of biotic stresses, including immune system, we can also conclude that the capsule is not only about virulence. However, what we really test is how the capsule influences generic process of adaptation to these different and novel environments.

2) Perhaps this is a matter of terminology, but I found it confusing to speak of reduced capsular expression in a strain lacking a capsule. It would help to more clearly distinguish the capsule itself from capsular genes.

We have checked the text for instances in which this could be confused and have clarified using the terminology “surface-associated polysaccharides”, as explained in the legend of figure 1.

3) There is discussion of “compensating” for the absence of a capsule, but it isn't clear what they are compensating for. Is this referring to fitness or something else? Some non-capsular strains produce excess polysaccharide. Is there any evidence that this recovers functions once served by the capsule?

We agree with this comment. We do not test specifically whether excess of polysaccharide recovers capsule functions. We have modified the text to avoid references to “compensation”.

4) Yield increases in populations without a capsule, but it isn't clear if the non-capsulated strains have higher yield than their WT parents to begin with. My naïve prediction would be that that capsule is costly, and so yield would start higher in these strains. Further degradation of a non-functional pathway would then be beneficial.

The capsule is not always costly, and this is strongly dependent in the environment. In our previous experiments (Buffet et al, 2021, Proc R Soc B) in well-mixed media, we showed that the capsule is costly in some media, like LB and ASM, but it provides fitness advantages and increases population yield in M02 and AUM. In spite of minor differences across strains, this is confirmed by the novel analyses of total population yield of each ancestor in each evolutionary condition (see new Figure S1). For example, the yield of capsulated bacteria in AUM is systematically higher than that of the respective non-capsulated mutants. Multifactorial ANOVA showed that the environment and ancestral strain affected yield significantly, but not the capsule genotype (Environment $F = 1164$, $df = 4$, $P < 0.001$; Ancestral Strain $F = 40.9$, $df = 2$, $P < 0.001$; Capsule Genotype $F = 0.001$, $df = 1$, $P = 0.9$) however there are significant interactions between ancestral and capsule genotype ($P < 0.001$) and genotype and environment ($P = 0.02$).

The paper notes that these traits evolved by co-evolution. I think this is better referred to as trait correlations and pleiotropy—1 change affecting several phenotypes and not two traits evolving in concert.

We have also removed references to coevolution.

5) Trait categories in Fig 2A should be spelled out rather than using the COG categories. This should at least be done for the categories where differences are observed.

This has been modified.

6) I understand the need to distinguish KlebEvo 1 and KlebEvoII internally, but I found the designation unnecessary and distracting in this paper. It's easier to say “we previously found...”, etc.

We have eliminated this distinction, following the reviewer's suggestion.

7) From Line 244 the paper looks at mutations leading to capsular loss. These are interesting data, but the justification is unclear (as is the interpretation of the comparison) if the expectation is that structure matters for phenotype and their causal mutations. The conclusion on line 271 is therefore somewhat vague “Taken together, our results show that, independently of the environmental structure, Klebsiella strains evolve and mutate by different mechanisms”. This would benefit from some elaboration.

We have streamlined this section. We now clearly outline the three different questions we raise. First, we aim to test whether the structure could influence the mutational mechanisms (*ie* types of mutations), by comparing the three focal strains in well mixed vs structured environment. Second, we expand our analyses to test whether the observed differences in capsule inactivation across these three strains or genetic backgrounds were similar to those observed in a larger Klebsiella diversity. Third, we address whether mutation mechanisms in the capsule operon are similar to those observed outside the capsule operon.

8) Social arguments/bet-hedging/etc are not well supported with data. There is an assumption that evolved strains have increased population structure, more social interactions, and that this affects cooperation. These are really interesting, and perhaps likely, possibilities. I hope they will be tested in future work with these strains, b

We agree with the reviewer. However, hypermucoidity in capsulated populations, and more biofilm production (most likely by the action of the mutated type 3 fimbriae) both result increased population structure. We will most definitely follow up on this. We have toned down our claims, but we decided to still include this, as we believe, and the reviewer recognizes, it is a likely possibility and thus can be legitimately discussed.

Reviewer #2 (Remarks to the Author):

Dear Authors,

I enjoyed reading your paper and I think it makes an important contribution to our understanding of how Kleb capsule positive and capsule negative variants adapt to novel environments. I have a couple of comments that largely relate to clarity of meaning regarding the text plus additional comments on statistics reporting, sequence analysis and discussion.

We thank the reviewer for the corrections and the helpful suggestions which have been incorporated to the text.

1. Grammar and syntax

Throughout the text the “adaptation to” and the “first steps in adaptation” is missing the qualifying words which is “novel environments”? They are not just randomly adapting to nothing and as such I think that you need to be clear throughout the text as to what they adapting to both in the general sense of the paper i.e. novel environments and then with regard to specific contexts i.e. soil, ASM etc...

a. As an example : Line 27 adaptation primarily occurs.... I think that this should read adaptation to novel environments and line 69, 429

We agree with reviewer #2, we have specified to what these strains are adapting to.

b. Lines 40-43 This sentence is very long and confusing, it needs to be broken down and/or rephrased

This has been rephrased.

c. Line 45 how components at the cell surface.... This is very vague, what are you referring to, what are the cell surface components?

This has been specified.

d. End of sentence in line 55 – this is a statement of fact and needs a reference

We have now added the appropriate reference “Rendueles et al. 2018 *PloS Genetics*”

e. Lines 56 to 59 starting at Metagenomic analysis... I think that this is a little bit confusing and could do with be better qualified and referenced.

We have rewritten this section in order to clarify it.

We have modified the following as suggested by the reviewer:

Line 60 change todiverse animals, but it is also found in.....

Line 62 needs a reference

Line 68 change to strongly influences the

Line 103 (HMP)

Line 105 spell out CFU

Line 308 Qualify your argument more clearly.... i.e. It is a very costly phenotype as measured by growth rate analysis....

Line 365 No study has....

f. Line 367 to 368 This sentence is both confusing and overstated.... You looked a number of phenotypic traits not all phenotypic traits possible... I would change this toHere, we show that the mechanisms of adaptation, as well as the associated genetic and phenotypic changes and traits measured in our study, depend on the presence of capsule.

We agree and we have now toned down our claims.

g. Line 517 to 519 I found this sentence very confusing, it is not very clear to me what days you were actually plating at ...eg. Every second day during ten days???

We plated all capsulated populations at day 2,4,6,8 and 10. This is now clearly spelled out in the methods section.

h. Line 544 – did you sequence 180 isolates? I would but the n = X here.

Yes, we sequenced one clone per population, and left out those for which contaminations were detected. N = 164. This has been precisely stated.

2. Statistics

a. I would like to see a full breakdown of how each statistical test in the methods not just the package that was used to calculate them. i.e. We used Fisher's Exact to test for X, Y and Z..... Two-way ANOVA for X, Y and Z.... and so on.

This has now been carefully clarified as suggested.

b. For all statistical analysis reported in the results and text please report in full e.g for ANOVA give F statistic, df and p.

For the multifactorial ANOVA testing for effects of environment, ancestor and capsule on the evolution of the different traits, F statistics and df are provided in Table S2 but excluded from the text for simplicity reasons.

All others are now embedded in the text, where relevant. For Fisher tests, odds ratio is provided.

c. I am not familiar with the inclusion of N, K and M prefixes before reporting the Fisher's test result? Also prefixes are missing for the reporting of this stat elsewhere

Following this comment, and reviewer #1, we modified the figure and define the different COG categories.

3. Other comments:

a. Line 120 Capsule production either remains near ancestral levels or is strongly diminished... could capsule production be at its maximum level of production?

The reviewer raises an interesting point. We would be inclined to think that the capsule is tightly regulated, but not at maximum level. Recent studies using transposon mutagenesis in two different strains (including Kpn NTUH used here) revealed that mutations can result in capsule hyperproduction (Dorman et al mBio, 2018. 9(6): p. e01863-18), suggesting that the strains are not at maximum level. We have included this idea in the discussion.

b. Figure 1. I found this figure very small and hard to read at the current size. Also please change the symbols for the strains as the colour for the two Kpn strains looks the same to me (even if the symbol is a different shape it is important to make them highly distinguishable)

The color has been changed and the figure resized.

c. Line 334 Our findings that HMP....without biotic pressure....

This is very interesting but I think it also raises an important point – in natural environments Kleb strains will be embedded in complex microbial communities (other bac, phages etc) and therefore the evolution of costly traits may be constrained if there is competition etc. Whilst experimental evo is very important to hypothesis testing it often uses highly reductionist experimental setups as you outline in this paper and whilst your experimental setup is not an issue (and the use of multiple abiotic environments is great), and does not preclude its acceptance, I think it is important to elaborate on the importance of community context in the evolution of the specific traits you measure (growth/pop size) and if you would anticipate similar results if there was e.g. a background community present in the discussion section.

This is a great commentary and we absolutely agree with the reviewer. We have now included this in the discussion. Preliminary experiments performed in our laboratory in which dual-species “communities” were evolved in ASM and AUM also revealed the emergence of

mucoviscosity in mixed communities (*NB*: this will not be mentioned in the manuscript, as data is still preliminary)

d. Sequence analysis – I have used Breseq and found that it generated several false positives and also failed to identify actual mutations when using the default setting. I would recommend complementing your analysis with Samtools or another programme. Also Breseq is primarily useful for indels and SNPs but I see that loss of plasmids were reported in Figure 4 - how were these detected?

We also used *snippy* (<https://github.com/tseemann/snippy>) to detect SNPs, but failed to mention it. We have now included this in the methods. The results from *snippy* support SNP calling by *breseq*. Further, prior to generating *wzc* and *ramA* mutants, the presence of these mutations were verified in the evolved clone by Sanger sequencing, and all were *bona fide* SNPs. This has been included in the methods.

Loss of plasmid is rare, and this was verified by the absence of reads mapping the plasmid.

Reviewer #3 (Remarks to the Author):

This is an interesting piece of work examining the evolution and adaptation of capsulated bacteria under several different conditions, and attempting to draw general principles that drive selection of certain traits that may or may not impact virulence. Its premise and setup of experiments are innovative and derive a lot from prior work of the same authors, further sharpening their arguments and providing more evidence. Deciding to examine capsule + and null strains in the evolution experiment is a very good decision and indeed the authors find that such background contributes to the evolution of different mutations that seems to be selected for different traits. The selection of traits to be examined is also very sound.

We thank the reviewer for the positive comments.

However, there are some points that need to be further clarified and addressed.

1. It is unclear what spatially structured environment means. Do the authors mean shaking versus no shaking? Or in fact means specific kinds of media? Why would eg AUM be considered spatially structured versus LB?

We mean shaking versus non shaking. All tested media are considered to be equally structured. This is now clearly specified in the methods section and the first time it is mentioned in the text.

2. It is unclear why cfu cannot be measured for HMP strains. How is cfu determined? Is it not by plating? It is not explained in methods. Or are authors referring to Fig 3 on deciding whether it is capsulated or uncapsulated?

HMP strains are highly dense, glued-together populations (see picture in Figure 7) that are very difficult to pipet, and cannot be easily resuspended and homogenized. Thus, impossible to accurately perform serial dilutions. Practically speaking, these populations form a *blob*, and either the whole *blob* is passed on to the next dilution, or almost sterile media is passed to the next dilution. This is now clearly explained in a new methods section.

3. Why can't HMP be measured for those evolved in LB? Could not the HMP assay be modified for differential centrifugation rather than reading OD of unpelleted vs pelleted bacteria in the original media? These data may be quite interesting to give more context to the evolution of traits.

We believe there is a confusion. We assessed the HMP phenotype in all populations, including those evolved in LB. Exceptionally, we kept the environment constant (in M02) for this experiment because results vary significantly across environments, as the environment itself can have different viscosities (*ie* ASM is composed of mucus, which already has a different density).

4. How is increased surface polysaccharides in non capsulated strains measured?

The same protocol was undertaken as to measure capsule production. To measure cell-bound surface polysaccharides, either in capsulated or non-capsulated backgrounds, cells are centrifuged, and the supernatant discarded. We then specifically measured the glucuronic acid present in the polymers that remained attached to the cell surface. This is now clarified in the methods section.

5. Figures 1-3, there are big differences between strains BJ1 and NTUH. The authors did not explain nor discuss about these 2 strains.

We first tested if the two strains evolved similarly with respect to their capsule genotype. In six independent trait x environments combinations, BJ1 and NTUH showed similar patterns (either their respective Cap⁺ and Cap⁻ evolved in the same direction or Cap⁺ and Cap⁻ evolved in opposite directions in both strains). We also observed that in 5 different trait x environment combinations, both strains evolved differently in respect to their capsule. The latter was mainly observed in M02. Specifically, capsulated BJ1 had significantly higher yield and surface polysaccharide production than the non-capsulated, but the opposite is true for NTUH, in M02.

We performed *post hoc* analyses to compare across strains (Multifactorial ANOVA, followed by Tukey's Honest Significant Difference test), and we observed that there are significant differences between the two *K. pneumoniae* strains in biofilm formation (P<0.001), surface polysaccharide production (P<0.001), and HMV (P =0.02), but not in population yield (P= 0.18).

We now mention this briefly in the results section.

For differences at the genetic level, a dissimilarity matrix of mutated genes across different ancestors is presented in Table S3.

6. Figure 5 is fascinating. In the *wzc* mutated evolved strains, did the authors check NTUH *rmpA* mutant would then give a more similar phenotype as the other strains showing evolved HMP in all media? Does the absence of *rmpA* or *rcs* gives a different ratio?

Our focal strain Kva342 does not have an *rmpA* allele, and its capsule production is not affected by *rscB*.

In Kpn BJ1 and Kpn NTUH, we did not check for the impact of *rsc* or *rmpA* mutations, but we expect it to be low, as the HMP was similar across clones with different *rmpA/rsc* genotypes, for example between clone 1C2 vs clones 3A1,3A2,6A5 from Kpn NTUH, Figure S6.

7. I find figure 6C very hard to understand. Can the authors think of a better way to represent the data? Confusing to know the letters represent clones and mutations in which genes.

We have modified this to clarify the figure.

8. The inability to reverse phenotypes or effect phenotypes in the *wzc* and *ramA* in Figure 6 for antibiotic susceptibility is a little disturbing. Can the authors examine their evolution

expts in different timepoints to see indeed *wzc* appear first before *ramA*? Or vice versa? And complement the change singly to test antibiotic susceptibility?

We believe there has been a misunderstanding due to the fact that our previous Figure 6C was unclear (see comment above). We specifically show in Figure 6C that insertion of a mutated/evolved *ramA* allele always results in increased sensitivity to antibiotics, independently of the *wzc* genotype. We show this by inserting the two evolved alleles in an ancestral background, and by reverting the two evolved alleles to ancestral in an evolved clone. We control these results by using double recombinant clones of each mutant, one with the ancestral allele and one with the evolved allele.

Further our analyses to test whether these mutations co-evolved (L409-L414) show that the emergence of mutations in *wzc* is independent of those in *ramA*, we thus feel it is not necessary to investigate which mutation appeared first.

9. It is unclear whether the authors examined growth rates of mutations seen in figure 6 as a reduced growth rates could result in antibiotic susceptibility as measured by MIC, being a more tolerant phenotype.

We have now measured the growth of ancestors, evolved clones and the generated *ramA* mutants. As expected, some evolved clones have a delayed growth rate in liquid. This is mostly due to mutations in *wzc* (as shown in the manuscript). However, insertion of evolved *ramA* alleles in the ancestral background does not result in any growth alteration in a capsulated background. This allows us to conclude that increased antibiotic sensitivity is not due to changes in growth. Interestingly, we do observe a delayed growth when *ramA* evolved allele is introduced in the non-capsulated ancestor. We speculate this is due to membrane instability. However, delayed growth should result in increased antibiotic tolerance, yet, here we observe increased sensitivity of the mutant. This is now shown in the new Figure S9 and mentioned in the appropriate results section.

Reviewers' Comments:

Reviewer #1:

Remarks to the Author:

The authors have done a very thorough job revising the manuscript in light of my comments and those of the other referees. Overall, this is really nice study that answers many questions and poses many more. I'll look forward to seeing the follow-up work in the future.

Reviewer #2:

Remarks to the Author:

The authors have addressed all the concerns I raised in their revisions and I think that the manuscript is now a more nuanced and balanced piece of work.

Reviewer #3:

Remarks to the Author:

The authors have significantly improved the clarity of their manuscript both in writing and figure representation, and better explanation in the methods. I do not have further comments for the authors to address.